# Few-Shot Diffusion Models Escape the Curse of Dimensionality

**Ruofeng Yang**[1], **Bo Jiang**[1], **Cheng Chen**[2], **Ruinan Jin**[34],
**Baoxiang Wang**[34], **Shuai Li**[*1]
[1] John Hopcroft Center for Computer Science, Shanghai Jiao Tong University
[2] East China Normal University
[3] The Chinese University of Hong Kong, Shenzhen    [4] Vector Institute
{wanshuiyin, bjiang, shuaili8}@sjtu.edu.cn,
chchen@sei.ecnu.edu.cn, {jinruinan,bxiangwang}@cuhk.edu.cn

## Abstract

While diffusion models have demonstrated impressive performance, there is a growing need for generating samples tailored to specific user-defined concepts. The customized requirements promote the development of few-shot diffusion models, which use limited $n_{ta}$ target samples to fine-tune a pre-trained diffusion model trained on $n_s$ source samples. Despite the empirical success, no theoretical work specifically analyzes few-shot diffusion models. Moreover, the existing results for diffusion models without a fine-tuning phase can not explain why few-shot models generate great samples due to the curse of dimensionality. In this work, we analyze few-shot diffusion models under a linear structure distribution with a latent dimension $d$. From the approximation perspective, we prove that few-shot models have a $\widetilde{O}(n_s^{-2/d} + n_{ta}^{-1/2})$ bound to approximate the target score function, which is better than $n_{ta}^{-2/d}$ results. From the optimization perspective, we consider a latent Gaussian special case and prove that the optimization problem has a closed-form minimizer. This means few-shot models can directly obtain an approximated minimizer without a complex optimization process. Furthermore, we also provide the accuracy bound $\widetilde{O}(1/n_{ta} + 1/\sqrt{n_s})$ for the empirical solution, which still has better dependence on $n_{ta}$ compared to $n_s$. The results of the real-world experiments also show that the models obtained by only fine-tuning the encoder and decoder specific to the target distribution can produce novel images with the target feature, which supports our theoretical results.

## 1 Introduction

In recent years, diffusion models have shown an excellent ability to generate diverse, high-quality samples and show state-of-the-art performance in many areas with large-scale, standard datasets (Rombach et al., 2022; Ho et al., 2022; Li et al., 2024a; Blattmann et al., 2023; Li et al., 2023b; Liu et al., 2024; Li et al., 2024b). However, users often desire to generate samples that resemble the ones they provide, such as images related to their families, daily lives, or specific items. These user-provided samples are typically limited in number and do not appear frequently in large-scale datasets. Consequently, training a diffusion model from scratch using such limited, personalized samples often results in poor performance. To cater the customized requirements of users, few-shot diffusion models attract much attention. Few-shot diffusion models aim to fine-tune a pre-trained diffusion model using a limited amount of data ($5 \sim 10$ samples), and they have recently delivered

---

[*]Corresponding author

impressive results in various domains, including image generation (Ruiz et al., 2023; Han et al., 2023; Zhu et al., 2023), video generation (Chen et al., 2023b), and the medical domain (Dutt et al., 2023).

Before the fine-tuning phase, we first need to train a diffusion model on the large source dataset $\{X_{s,i}\}_{i=1}^{n_s}$ as the pre-trained model. A diffusion model consists of a forward process and a reverse process (Song et al., 2020). The forward process gradually converts the data distribution into Gaussian noise. The reverse process sequentially removes the noise in the data to generate samples, which relies on the gradient of logarithmic forward process density (a.k.a. score function). To run the reverse process, diffusion models use a neural network to approximate the unknown score function.

With a pre-trained diffusion model, the paradigm to obtain a few-shot diffusion model is to fine-tune the model using a limited target dataset $\{X_{ta,i}\}_{i=1}^{n_{ta}}$. In earlier times, fully fine-tuned methods, such as DreamBooth (Ruiz et al., 2023), provided an important boost for developing few-shot models. However, they also show that the diffusion models suffer from the overfitting and memory phenomenon when fine-tuning all parameters. Furthermore, a fully fine-tuned method is both memory and time inefficient (Xiang et al., 2023). To avoid the above problems, many works freeze most parameters and fine-tune some key parameters, such as cross-attention layers (Kumari et al., 2023; Moon et al., 2022), some concept neurons (Liu et al., 2023) or text-embedding (Gal et al., 2022), to approximate the ground-truth target score function. These works not only preserve the prior information but also have a lower requirement for the target dataset size, which is more practical for applications. Hence, we aim to explain the great performance of these models in this work.

Despite the empirical success, no existing theoretical work specifically analyzes the approximation bound for few-shot diffusion models, and the following question remains open:

*Do few-shot diffusion models with a fine-tuning phase enjoy a small approximation error with a limited target dataset?*

For the approximation error bound, some works currently analyze diffusion models without a fine-tuning phase (Oko et al., 2023; Chen et al., 2023c; Yuan et al., 2023; Li et al., 2023c). Importantly, when analyzing general, bounded data, these works suffer from the curse of dimensionality. More specifically, Oko et al. (2023) analyze bounded distribution and show the $n_s^{-s'/D}$ approximation bound, where $D$ is the data dimension of $X_s$. Chen et al. (2023c) analyze linear structure distribution $X_s = A_s Z$ with subgaussian latent variable $Z \in \mathbb{R}^d$ and achieve $n_s^{-2/d}$ results. Since the source dataset size is large enough, the influence of dimension is tolerable. However, for the limited target dataset, if trivially using the above technique, the bound is $n_{ta}^{-1/D}$ or $n_{ta}^{-2/d}$, which is large and can not explain why few-shot diffusion models efficiently approximate the target score function.

In this work, for the first time, we propose the approximation bound specifically to few-shot diffusion models with a fine-tuning phase and prove that the few-shot diffusion model can escape the curse of dimensionality. More specifically, we show that when assuming (1) linear structure data and (2) the source and the target data share latent distribution, the few-shot diffusion models with a fine-tuning phase achieve $\widetilde{O}(n_s^{-2/d} + n_{ta}^{-1/2})$ approximation error bound, which makes the first step to explain why few-shot diffusion models have great performance in the application. Generally speaking, due to the component $n_{ta}^{-1/2}$, the few-shot diffusion only needs a few target samples to achieve the same bound compared to $n_s^{-2/d}$. To support our augmentation, we calculate the requirement of $n_{ta}$ to obtain an accurate enough approximated target score function in popular datasets. Table 1 shows that the requirement of $n_{ta}$ is about $5 \sim 10$ samples, matching the customized diffusion model requirement. We also do experiments on the real-world datasets and show that 10 target images are enough for few-shot models to generate novel images with the target feature (see Section 6).

After directly using the property of the minimizer to obtain an approximation bound, we analyze how to optimize the few-shot diffusion models to obtain a minimizer. Since the score-matching objective function is highly non-convex, only a few works analyze the optimization problem of diffusion models (Shah et al., 2023; Bruno et al., 2023; Cui et al., 2023; Li et al., 2023c). Furthermore, these works either require (1) an exponential size neural network (Li et al., 2023c) or (2) a distribution determined by one variable (Shah et al., 2023; Bruno et al., 2023; Cui et al., 2023) to simplify the optimization problem. This work proves that few-shot diffusion models can simplify the optimization problem without these requirements. When analyzing the optimization problem, we focus on a

Gaussian latent variable special case [2]. Then, we prove that the expected few-shot objective function has a closed-form minimizer, which means the empirical solution can be directly obtained without a complex optimization process. We also prove the accuracy bound $\widetilde{O}(1/n_{ta} + 1/\sqrt{n_s})$ of empirical closed-form solution, which still has better dependence on the target dataset. In conclusion, we accomplish the following results for few-shot diffusion models under linear structure distribution:

- For the approximation bound, we consider a subgaussian latent variable and prove $\widetilde{O}(n_s^{-2/d} + n_{ta}^{-1/2})$ bound for few-shot models, which is better than $n_{ta}^{-2/d}$ result without fine-tuning.

- For the optimization problem, we consider a latent Gaussian special case and prove that the expected few-shot objective function has a closed-form minimizer. Furthermore, we prove the accuracy bound $\widetilde{O}(1/n_{ta} + 1/\sqrt{n_s})$ for the empirical closed-form solution.

- To support our theoretical results, we do real-world experiments and show that the models obtained by only fine-tuning specific encoder and decoder can use only 10 target images to generate novel images with the target feature.

## 2 Related Work

**The approximation error bound.** Recently, some works analyze the approximation error bound of diffusion models without a fine-tuning phase. Oko et al. (2023) analyze $s'$-order bounded derivatives distribution and show the approximation error bound is $n_s^{-s'/D}$. Chen et al. (2023c) analyze distribution with linear structure and subgaussian latent variable and show that the $n_s^{-2/d}$ result. The approximation error bound of the above works suffers the curse of (latent) dimensionality. To avoid this phenomenon, some works analyze special data distributions. Shah et al. (2023) and Cui et al. (2023) analyze the mixture of Gaussian (MOG) with known variance and achieve a $1/n_s$ approximation bound. Yuan et al. (2023) analyze linear structure distribution with Gaussian latent variable and achieve $1/\sqrt{n_s}$ result. Mei & Wu (2023) analyze Ising models and prove that the term corresponds to $n_s$ is $1/\sqrt{n_s}$. However, the remaining terms do not converge to $0$ when $n_s$ goes to $+\infty$. For general bounded data distribution, Li et al. (2023c) provide a $n_s^{-2/5}$ approximation error bound. However, they use a 2-layer random feature network and only allow the second linear layer to be trainable. Hence, the network size is $\exp(n_s)$ compared to $\text{Poly}(n_s)$ size of all previous works.

**The optimization of diffusion models.** Since the score matching objective function is highly non-convex, only a few works analyze how to optimize it to obtain a minimizer (Shah et al., 2023; Cui et al., 2023; Bruno et al., 2023; Li et al., 2023c). These works either make assumptions about data distribution or network size to guarantee only one optimization variable, leading to a simpler optimization problem. For special data distributions, Bruno et al. (2023) and Cui et al. (2023) analyze a Gaussian with fixed variance and a 2-mode mixture of Gaussian (MOG) with equal, trainable mean and fixed variance, respectively. Shah et al. (2023) analyze a multi-mode MOG with a fixed variance and prove a local convergence guarantee. Since they assume the distance between any two modes is large enough and a good enough initialization, the optimization problem is similar to optimizing a Gaussian distribution. For the large neural network size, Li et al. (2023c) analyze a general, bounded distribution with a 2-layer NN. Note that they require $\exp(n_s)$ hidden neurons and only allow the linear layer to be trainable, which also leads the optimization problem to a convex optimization.

## 3 The Introduction of Few-shot Diffusion Models

With pre-trained models, the paradigm to obtain a few-shot diffusion model is to freeze most parameters and fine-tune some key parameters corresponding to the target data distribution. Since the analysis of few-shot diffusion models relies heavily on the pre-trained model, this section first provides a concise overview of the fundamental concepts and notations associated with diffusion models. Then, we introduce the paradigm of few-shot diffusion models in Section 3.2.

---

[2]Though it is a special case, the previous analysis can not be used since it is determined by two components.

## 3.1 The Forward and Reverse Process

Let $q_0$ be the data distribution. Given $X_0 \sim q_0 \in \mathbb{R}^D$, non-decreasing function $f(X_t, t)$ and $g(t)$, the forward process is defined by:

$$\mathrm{d}X_t = f(X_t, t)\mathrm{d}t + g(t)\mathrm{d}B_t \,,$$

where $\{B_t\}_{t \in [0,T]}$ is a $D$-dimensional Brownian motion. In this work, we choose $f(X_t, t) = -1/2 X_t$ and $g(t) = 1$, which corresponds to variance preserving (VP) forward process and is widely used in practice [3](Shah et al., 2023; Song et al., 2020). Let $q_t$ be the density function of $X_t$. Once a forward process is chosen, the conditional distribution of $X_t | X_0$ is $q_t(X_t | X_0) = \mathcal{N}(m_t X_0, \sigma_t^2 I_D)$, where $m_t = e^{-t/2}, \sigma_t^2 = 1 - e^{-t}$. Note that when $t$ goes to $+\infty$, $q_t$ converges to $\mathcal{N}(0, I_D)$, which is helpful in choosing the initial distribution for the sampling process.

To generate samples, diffusion models reverse the forward SDE and run the reverse process. Since the reverse process contains the gradient of forward logarithmic density $\nabla \log q_t(\cdot)$ (a.k.a. score function), the model approximates it by using a neural network $\mathbf{s}(\cdot, t)$ and the score matching objective function (see Section 3.2). After that, diffusion models discretize the continuous reverse process to obtain an implementable algorithm. Let $t_0 \leq t_1 \leq \cdots \leq t_K = T$ be the discretization points in the forward time and $h_k = t_k - t_{k-1}$ be the $k$-th stepsize. When considering the reverse time, we define $t'_k = T - t_{K-k}$. In this work, we choose the exponential integrator (EI) discretization scheme, which has great performance (Zhang & Chen, 2022). The EI discretization freezes the approximated score at $t'_k$ and runs the following process in the reverse time:

$$\mathrm{d}\widehat{Y}_t = \left[ f(\widehat{Y}_t, T - t) + g(T - t)^2 \mathbf{s}(\widehat{Y}_{t'_k}, T - t'_k) \right] \mathrm{d}t + g(T - t)\mathrm{d}B_t \,, t \in [t'_k, t'_{k+1}] \,,$$

where $\widehat{Y}_0 \sim \mathcal{N}(0, I_D)$ due to the stationary distribution of the forward process.

While the discretization complexity $K$ has been well-studied with an accurate enough score function (Benton et al., 2023; Li et al., 2023a), there is a lack of analysis for the score-matching process. Therefore, this work focuses on the score approximation and the optimization problem of the few-shot score-matching objective function.

## 3.2 The Score Matching Objective Function

In this work, we specifically analyze few-shot diffusion models, which involve two datasets: (1) the source dataset $\{X_{s,i}\}_{i=1}^{n_s}$; (2) the target dataset $\{X_{ta,i}\}_{i=1}^{n_{ta}}$. The approach involves first training a pre-trained diffusion model on the source dataset and then freezing the backbone network to fine-tune the diffusion models on the target dataset.

For data distributions, we assume that the source distribution $q^s$ and the target distribution $q^{ta}$ are both supported on a low-dimensional linear subspace. The low-dimensional structures have been discovered on many popular image datasets (Pope et al., 2021; Gong et al., 2019; Tenenbaum et al., 2000) due to the locally connected and symmetrical property, and it is crucial for diffusion models. For image generation, current popular diffusion models, such as Stable Diffusion (Rombach et al., 2022), transform images to a latent space and run diffusion models in the latent space. Except for the image generation, Chen et al. (2024) recently show the latent dimension plays an important role in diffusion models to work well in self-supervised learning, and linear subspace is enough.

We further assume that the source and target data share the same latent distribution. Note that this is a common assumption in few-shot learning. In particular, previous theoretical works in the context of supervised few-shot learning often assume that the source and target distributions have a common latent representation (Du et al., 2020; Chua et al., 2021; Meunier et al., 2023).

**Assumption 3.1.** The source datapoints $X_s$ and target datapoint $X_{ta}$ admit a low dimensional linear structure and share the same latent distribution $X_s = A_s Z$ and $X_{ta} = A_{ta} Z$ where $A_s, A_{ta} \in \mathbb{R}^{D \times d}$ with orthonormal columns and $Z \sim q_z \in \mathbb{R}^d$.

---

[3]Our analysis can be extended to $f(X_t, t) = -1/2\beta_t X_t$ and $g(t) = \sqrt{\beta_t}$, where $\{\beta_t\}_{t \geq 0}$ is non-decreasing and bounded sequence.

As mentioned in Chen et al. (2023c), when assuming linear distribution, the ground-truth score function is decomposed into the latent score function $\nabla \log q_t^{\mathrm{LD}}(Z')$ and linear encoder and decoder:

$$\nabla \log q_t^s(X) = A_s \nabla \log q_t^{\mathrm{LD}}\left(A_s^\top X\right) - \frac{1}{\sigma_t^2}\left(I_D - A_s A_s^\top\right) X\,,$$

where $q_t^{\mathrm{LD}}(Z') = \int q_t(Z'|Z)\, q_z(Z)\mathrm{d}Z$ and $q_t(\cdot|Z) = \mathcal{N}(m_t Z, \sigma_t^2 I_d)$. This form indicates that the diffusion process happens in the latent subspace. A conceptual way to approximate the score function is to minimize the following loss on a function class $\mathcal{S}_{NN}$:

$$\min_{s \in \mathcal{S}_{NN}} \int_0^T w(t)\mathbb{E}_{X_t \sim q_t^s}\left\|\nabla \log q_t^s\left(X_t\right) - \mathbf{s}\left(X_t, t\right)\right\|_2^2 \mathrm{d}t\,,$$

where $w(t)$ is a weight function. However, the above objective function is intractable since $\nabla \log q_t(\cdot)$ is unknown. Vincent (2011) propose the following implementable loss:

$$\mathcal{L}_s(\mathbf{s}) = \int_0^T w(t)\mathbb{E}_{X_0}\left[\mathbb{E}_{X_t|X_0}\left\|\nabla \log q_t^s\left(X_t|X_0\right) - \mathbf{s}\left(X_t, t\right)\right\|_2^2\right]\mathrm{d}t\,.$$

Due to the forward process, $\nabla \log q_t^s\left(X_t|X_0\right)$ has an analytical form and is equal to $-(X_t - m_t X_0)/\sigma_t^2$. Vincent (2011) also prove that this objective function only has a constant difference compared to the above one. The empirical loss with the source datasets $\{X_{s,i}\}_{i=1}^{n_s}$ is defined by:

$$\min_{\mathbf{s}_{V,\theta} \in \mathcal{S}_{NN}} \widehat{\mathcal{L}}_s(\mathbf{s}_{V,\theta}) = \frac{1}{n_s(T-\delta)} \sum_{i=1}^{n_s} \int_\delta^T \ell_t^s\left(X_{s,i}; \mathbf{s}_{V,\theta}\right)\mathrm{d}t\,, \tag{1}$$

where

$$\ell_t^s\left(X_{s,i}; \mathbf{s}\right) = \mathbb{E}_{X_t|X_0=X_{s,i}}\left[\left\|\nabla \log q_t^s\left(X_t|X_0\right) - \mathbf{s}\left(X_t, t\right)\right\|_2^2\right]\,,$$

and

$$\mathcal{S}_{\mathrm{NN}} = \left\{\mathbf{s}_{V,\boldsymbol{\theta}}(X,t) = \frac{1}{\sigma_t^2}V\mathbf{f}_\theta\left(V^\top X, t\right) - \frac{1}{\sigma_t^2}X : V \in \mathbb{R}^{D \times d} \text{ with orthonormal columns,}\right.$$

$$\left.\mathbf{f}_\theta : \mathbb{R}^d \times [\delta, T] \to \mathbb{R}^d \text{ a ReLU network}\right\}\,.$$

Note that we take $w(t) = 1/(T-\delta)$ for simplicity, where $\delta$ is the early stopping parameter to avoid the blow-up phenomenon of score functions at the end of reverse process. Furthermore, we take the integral over the forward time instead of discretizing the timeline since $X_t$ is easy to generate.

The linear encoder and decoder structure and the shortcut connection in $\mathcal{S}_{NN}$ is due to the form of the ground-truth score function. The specific parameters for $\mathbf{f}_\theta$, such as its length and width, are identical to those used in Chen et al. (2023c). Generally, with a given network accuracy parameters $\epsilon$, the network size is $\mathrm{Poly}(1/\epsilon)$. We show the parameter of neural network in Appendix A.

The diffusion models minimize the empirical loss to obtain a pre-trained approximated score function. Let the minimizer of Equation (1) be $(\widehat{V}_s, \widehat{\theta})$. Chen et al. (2023c) show that $(\widehat{V}_s, \widehat{\theta})$ leads a $n_s^{-2/d}$ approximation error bound. If trivially replacing $n_s$ with $n_{ta}$, we obtain a $n_{ta}^{-2/d}$ bound for the target dataset without the fine-tuning phase. Note that this bound suffers from the influence of the latent dimension $d$, which is still large in popular datasets (Table 1). In the next paragraph, we introduce the few-shot diffusion models with a fine-tuning phase and show that the dependence on $n_{ta}$ is $n_{ta}^{-1/2}$ in the error bound (Theorem 4.3).

**The Few shot Diffusion Models with a Fine-tuning Phase.** Since the source and target distribution share the same latent data distribution, we freeze $\hat{\theta}$ and only fine-tune the low-rank linear encoder and decoder layer $V_{ta}$. This method can significantly reduce the fine-tuning parameters and is similar to LoRA (Hu et al., 2021), which also fine-tunes two low-rank matrices and is widely used in fine-tuning the stable diffusion (Rombach et al., 2022).

Let $\ell_t^{ta}$ be the loss function of the target dataset at time $t$, which has similar definition compared to $\ell_t^s$. The optimization problem for the target dataset is

$$\min_{\mathbf{s}_{V_{ta},\hat{\theta}} \in \mathcal{Q}_{NN}(\hat{\theta})} \widehat{\mathcal{L}}_{ta}(\mathbf{s}_{V_{ta},\hat{\theta}}) = \frac{1}{n_{ta}(T-\delta)} \sum_{i=1}^{n_{ta}} \int_\delta^T \ell_t^{ta}\left(X_{ta,i}; \mathbf{s}_{V_{ta},\hat{\theta}}\right)\mathrm{d}t\,,$$

Table 1: The requirement of $n_{ta}$ in popular datasets. We use latent dimension in Pope et al. (2021).

| Dataset | CIFAR-10 | CIFAR-100 | CelebA | MS-COCO | ImageNet |
|---|---|---|---|---|---|
| Dataset Size | $6 \times 10^4$ | $6 \times 10^4$ | $2 \times 10^5$ | $3.3 \times 10^5$ | $1.2 \times 10^6$ |
| Latent Dimension | 25 | 22 | 24 | 37 | 43 |
| The Requirement of $n_{ta}$ | 6 | 8 | 8 | 5 | 5 |

where

$$\mathcal{Q}_{\mathrm{NN}}(\theta) = \left\{ \mathbf{s}_{V,\theta}(X,t) = \frac{1}{\sigma_t^2} V \mathbf{f}_\theta \left( V^\top X, t \right) - \frac{1}{\sigma_t^2} X : V \in \mathbb{R}^{D \times d} \text{ with orthonormal columns.} \right\},$$

Similarly, we define the minimizer of the few-shot objective function as $(\widehat{V}_{ta}, \widehat{\theta})$.

**Notations.** We denote by $I_D$ the $D$-dimensional identity matrix. For $X \in \mathbb{R}^D$ and $A \in \mathbb{R}^{D \times d}$, we denote by $\|X\|_2$ the Euclidean norm for vector and $\|A\|_F$ the Frobenius norm for matrix. We denote by $\|X\|_{L^2(q)}^2$ the expectation of $X$ in $L_2$ norm $\mathbb{E}_{X \sim q}[\|X\|_2^2]$.

## 4 Few-shot Diffusion Models Enjoy Better Approximation Error Bound

In this section, we show that few-shot diffusion models with a fine-tuning phase escape the curse of latent dimensionality and have a $\widetilde{O}(n_s^{-2/d} + n_{ta}^{-1/2})$ approximation bound [4]. This result makes the first step to explain why few-shot models have great performance with a limited target dataset.

Before showing our results, we first introduce standard assumptions on the latent distribution and the on-support ground-truth score function. We first assume that $Z$ has a subgaussian tail and the minimum eigenvalue of $Z$ is lower bound by $c_0$, also used in Chen et al. (2023c).

**Assumption 4.1.** $q_z > 0$ is twice continuously differentiable, $\lambda_{\min}(\mathbb{E}\left[ZZ^\top\right]) \geq c_0$ and $\mathbb{E}\|Z\|_2^2 \leq C_Z$. Moreover, there exist positive constants $B, C_1, C_2$ such that when $\|Z\|_2 \geq B$, $q_z(Z) \leq (2\pi)^{-d/2} C_1 \exp\left(-C_2 \|Z\|_2^2/2\right)$.

**Assumption 4.2.** The on-support ground truth score $A_s \nabla \log q_t^{\mathrm{LD}}(Z)$ and $A_{ta} \nabla \log q_t^{\mathrm{LD}}(Z)$ is $\beta$-Lipschitz in $Z \in \mathbb{R}^d$ for any $t \in [0, T]$.

Note that different from previous works directly assume $\nabla \log q_t(\cdot)$ is Lipschitz (Chen et al., 2022, 2023d), the $\beta$-Lipschitz on-support score function assumption does not conflict with the blow-up phenomenon when $t$ goes to 0 due to the existence of $(I_D - AA^\top)X/\sigma_t^2$. With these assumptions, we prove the approximation bound for few-shot models with a fine-tuning phase.

**Theorem 4.3.** Let $\alpha(n) = \frac{d \log \log n}{\log n}$, $F = \frac{(d + C_Z)d^2\beta^2}{\delta^2 c_0}$ and network parameter $\epsilon = n_{ta}^{-1/2}$. Assume Assumption 3.1, 4.1, 4.2 and $n_{ta}^{\frac{d+5}{4(1-\alpha(n_s))}} \geq n_s$. Then, with probability $1 - \delta_1$, the following inequality holds (hiding logarithmic factors)

$$\frac{1}{T - \delta} \int_\delta^T \left\| s_{\widehat{V}_{ta}, \widehat{\theta}}(\cdot, t) - \nabla \log q_t^{ta}(\cdot) \right\|_{L^2(q_t^{ta})}^2 \mathrm{d}t \leq \tilde{O}\left( \left( \frac{(1+\beta)^2 D d^3}{\delta (T - \delta) \sqrt{n_{ta}}} + F n_s^{-\frac{2 - 2\alpha(n_s)}{d+5}} \right) \log\left(\frac{1}{\delta_1}\right) \right).$$

The dependence of $\delta$ is due to the blow-up property of the score function. Note that when $n_s$ is sufficiently large, $\alpha(n_s)$ is negligible. Then, the approximation error bound for few-shot diffusion models is $\tilde{O}(n_s^{-2/d} + n_{ta}^{-1/2})$. Compared to the approximation error bound $n_{ta}^{-2/d}$ without a few-shot phase, it is clear that few-shot diffusion models escape the curse of the (latent) dimensionality.

*Remark* 4.4 (The discussion on the coefficient in Theorem 4.3). The goal of the fine-tuning phase is to achieve the same order error bound compared with the pre-trained models, which means that we consider the relative relationship between $n_{ta}$ and $n_s$. Hence, if the coefficient of $n_{ta}$ and $n_s$ has the same order, we can only consider $1/\sqrt{n_{ta}}$ and $n_s^{-2/d}$. To support the above augmentation, we calculate the coefficient of $n_s$ and $n_{ta}$ in detail. The dominated term of coefficient for $n_{ta}$ and $n_s$ are $Dd^3/\delta$ and $d^3/(\delta^2 c_0)$, respectively. The classic choice for the early stopping parameter $\delta$ and

---

[4] Here, the approximation error means the score matching error with finite source and target datasets.

forward time $T$ are $10^{-3}$ and 10, respectively (Karras et al., 2022). Then, with $D = 256 \times 256 \times 3$ as an example [5], $Dd^3/\delta = d^3 \times 20 \times 10^6$ and $d^3/(\delta^2 c_0) = d^3 \times 10^6/c_0$, which has the same order. Hence, we consider the relative relationship between $1/\sqrt{n_{ta}}$ and $n_s^{-2/d}$.

## 4.1 Discussion on the Approximation Bound

**The relationship to empirical phenomenon.** In applications, current few-shot diffusion models only require $5 \sim 10$ target images to achieve great performance. Theorem 4.3 makes the first step to explain why the few-shot diffusion models have great performance with a limited target $n_{ta}$. More specifically, with known source dataset size $n_s$ and the corresponding latent dimension $d$, we can calculate the inequality $n_{ta}^{\frac{d+5}{4(1-\alpha(n_s))}} \geq n_s$ [6] to obtain the requirement of $n_{ta}$ to achieve the same accuracy compared to the pre-trained diffusion models. Combined with the latent dimension of popular datasets (Pope et al., 2021), Table 1 shows the requirement of $n_{ta}$. It is clear that we only need less than 10 target images to obtain an accurate enough few-shot diffusion model that matches the performance in reality. The real-world experiments also support our discussion (Section 6).

Table 1 shows that the requirement of $n_{ta}$ is heavily influenced by the latent dimension $d$. When $d$ is large (e.g. ImageNet), the approximation bound of pre-trained models is influenced by latent dimension and has a large approximation error even with large-size source data. We only need a few target data to achieve the same error in this setting. When $d$ is small (e.g. CIFAR-10), pre-trained models have a small approximation error. We need a slightly larger target data size.

**The approximation error of the fully fine-tuned method.** As shown in our real experiment Section 6 and DreamBooth (Ruiz et al., 2023), when fine-tuning all parameters with a small target dataset, models tend to overfit and lose the prior information from the pre-trained model. In our theorem, this phenomenon means that in the fine-tuning phase, the model does not use $\widehat{\theta}$ learned by the pre-trained model and achieves a $n_{ta}^{-2/d}$ approximation error bound, which suffers from the curse of dimensionality. From an intuitive perspective, the probability density function (PDF) of a distribution learned by an overfitting model is only positive at the interval around the target dataset, which is far away from the PDF of true distribution and leads to a large error term. We also note that it is possible to avoid this phenomenon by using a specific loss (Ruiz et al., 2023) or carefully choosing the optimization epochs (Li et al., 2023c). We leave them as interesting future works.

**Proof sketch.** The first step is to prove that in $\mathcal{Q}_{NN}(\widehat{\theta})$, there exists a solution $(\bar{V}_{ta}, \widehat{\theta})$ has the following inequality (only focusing on $n_s$ and $n_{ta}$)

$$\frac{1}{T-\delta} \int_\delta^T \left\| \mathbf{s}_{\bar{V}_{ta},\widehat{\theta}}(X,t) - \nabla \log q_t^{ta}(X) \right\|_{L^2(q_t^{ta})}^2 \, dt \leq O\left( \epsilon^2 + n_s^{-\frac{2-2\alpha(n_s)}{d+5}} \log\left(\frac{1}{\delta_1}\right) \right).$$

To prove the above inequality, we first do the following decomposition:

$$\left\| \mathbf{s}_{\bar{V}_{ta},\bar{\theta}}(\cdot,t) - \nabla \log q_t^{ta}(\cdot) \right\|_2^2 + \left\| \mathbf{s}_{\bar{V}_{ta},\widehat{\theta}}(\cdot,t) - \mathbf{s}_{\bar{V}_{ta},\bar{\theta}}(\cdot,t) \right\|_2^2,$$

where $(\bar{V}_{ta}, \bar{\theta}) \in \mathcal{S}_{NN}$ is a constructed solution. The first term is due to the accuracy of the constructive neural network with network accuracy parameter $\epsilon$. For the second term, since the latent score function is shared and few-shot diffusion models directly use $\widehat{\theta}$, it is control by the approximation bound of the pre-trained diffusion models. Then, by using the inequality

$$\inf_{s_{V_{ta},\widehat{\theta}} \in \mathcal{Q}(\widehat{\theta})} \widehat{\mathcal{L}}_{ta}\left( \mathbf{s}_{V_{ta},\widehat{\theta}} \right) \leq \widehat{\mathcal{L}}_{ta}\left( \mathbf{s}_{\bar{V}_{ta},\widehat{\theta}} \right),$$

we build the bridge between $\mathbf{s}_{\widehat{V}_{ta},\widehat{\theta}}$ and $\mathbf{s}_{\bar{V}_{ta},\widehat{\theta}}$.

The second step is using the concentration to control the error between empirical $\widehat{\mathcal{L}}_{ta}$ and expected $\mathcal{L}_{ta}$. Roughly speaking, we have that

$$\mathcal{L}_{ta}\left( \mathbf{s}_{\widehat{V}_{ta},\widehat{\theta}} \right) - \widehat{\mathcal{L}}_{ta}\left( \mathbf{s}_{\widehat{V}_{ta},\widehat{\theta}} \right) \leq \frac{1}{n_{ta}\epsilon^2} \log\left( \mathcal{N}\left( 1/n_{ta}, \mathcal{Q}_{NN}(\widehat{\theta}), \|\cdot\|_2 \right) / \delta_1 \right),$$

---

[5] Since smaller $D$ is more friendly to $n_{ta}$, our discussion holds for all datasets in Table 1.

[6] This also indicates the requirement of $n_{ta}$ in Theorem 4.3 is easy to satisfy.

where $\mathcal{N}(1/n_{ta}, \mathcal{Q}_{NN}(\widehat{\theta}), \|\cdot\|_2)$ is the covering number of $\mathcal{Q}_{NN}(\widehat{\theta})$ in $L_2$ norm. Since only $V \in \mathbb{R}^{D \times d}$ can be optimized and $\widehat{\theta}$ is fixed in $\mathcal{Q}_{NN}(\widehat{\theta})$,

$$\log \left( \mathcal{N} \left( 1/n_{ta}, \mathcal{Q}_{\mathrm{NN}}(\widehat{\theta}), \|\cdot\|_2 \right) \right) = \widetilde{O}(Dd \log(1/n_{ta})).$$

Then, we balance different terms and achieve the final bound by choosing $\epsilon^2 = 1/\sqrt{n_{ta}}$.

## 5  The Few-shot Diffusion Model Have a Closed-form Minimizer

This section focuses on how to optimize the few-shot diffusion model. When considering the optimization problem, we assume the shared latent distribution admits an isotropic Gaussian distribution $q_z = \mathcal{N}(0, \lambda^2 I_d)$ with $\lambda^2 > 0$, which indicates the score function has the following formulation:

$$\nabla \log q_t^{ta}(X) = -\frac{1}{\lambda^2} A_{ta} A_{ta}^\top X - \frac{1}{\sigma_t^2} \left( I_D - A_{ta} A_{ta}^\top \right) X.$$

Note that though $q_z = \mathcal{N}(0, \lambda^2 I_d)$ is a special case of Assumption 4.1, we still need to know $\lambda^2$ and $A_{ta}$ to generate samples, which indicates the previous optimization analysis for diffusion models without a fine-tuning phase can not be used.

We fix a $t \in [\delta, T]$ for the few-shot objective function since the matrix $A_{ta}$ is independent of time $t$. More specifically, with an approximated latent distribution $\widehat{q}_z = \mathcal{N}(0, \widehat{\Sigma})$, where $\widehat{\Sigma} = \widehat{\lambda}^2 I_d$, the expected few-shot objective function at a fixed time $t$ is

$$\min_{\mathbf{s}_{V_{ta}, \widehat{\Sigma}} \in \widetilde{\mathcal{Q}}_{NN}(\widehat{\Sigma})} \mathcal{L}_{ta,t}(\mathbf{s}_{V_{ta}, \widehat{\Sigma}}) = \mathbb{E}_{X_{ta} \sim q^{ta}} \left[ \ell_t^{ta} \left( X_{ta}; \mathbf{s}_{V_{ta}, \widehat{\Sigma}} \right) \right].$$

where

$$\tilde{\mathcal{Q}}_{\mathrm{NN}}(\Sigma) = \left\{ \mathbf{s}_{V,\theta}(X, t) = \frac{1}{\sigma_t^2} V \mathbf{f}_\Sigma \left( V^\top X, t \right) - \frac{1}{\sigma_t^2} X : V \in \mathbb{R}^{D \times d} \text{ with } \mathrm{rank}(V) = d. \right\},$$

In this case, $\mathbf{f}_{\widehat{\Sigma}}(Z, t) = (I_d - \sigma_t^2 \widehat{\Sigma}_t^{-1}) Z$, where $\widehat{\Sigma}_t = m_t^2 \widehat{\Sigma} + \sigma_t^2 I_d$. The constraint $\mathrm{rank}(V) = d$ is used to guarantee that the few-shot diffusion models learn meaningful subspace instead of $\mathbf{0}^{D \times d}$. Note that $\mathrm{rank}(V) = d$ is a weaker constraint than $V^\top V = I_d$ since the pre-trained diffusion model has already learned the length information. This weaker constraint means we need less prior knowledge compared to $\mathcal{Q}(\theta)$, which is more user-friendly. Let $\widetilde{V}_{ta}$ be a minimizer of the above expected few-shot objective function. We show that $\widetilde{V}_{ta}$ has a closed form and good property.

**Lemma 5.1.** *Assume Assumption 3.1 and $q_z = \mathcal{N}(0, \lambda^2 I_d)$. Let $C = \mathbb{E}_{X_{ta} \sim q^{ta}} \left[ X_{ta} X_{ta}^\top \right]$ be the expected data covariance matrix. Then, $\widetilde{V}_{ta}$ has a closed form:*

$$\widetilde{V}_{ta} \widetilde{V}_{ta}^\top = \frac{m_t^2 \widehat{\lambda}^2 + \sigma_t^2}{\widehat{\lambda}^2} \left( C + \sigma_t^2 I_D \right)^{-1} C.$$

Lemma 5.1 indicates that few-shot diffusion models can directly obtain an approximation of the minimizer without a complex optimization process. Furthermore, this minimizer has good properties and exactly recovers the subspace spanned by $A_{ta}$. More specifically, the expected minimizer indicates $\|\widetilde{V}_{ta} \widetilde{V}_{ta}^\top - A_{ta} A_{ta}^\top\|_F^2 = 0$ when $n_s$ and $n_{ta}$ are infinite. However, the source datasets $n_s$ and target datasets $n_{ta}$ are finite, we analyze the empirical closed-form solution

$$\bar{\widetilde{V}}_{ta} \bar{\widetilde{V}}_{ta}^\top = \frac{m_t^2 \widehat{\lambda}^2 + \sigma_t^2}{\widehat{\lambda}^2} (m_t^2 \bar{C} + \sigma_t^2 I_D)^{-1} \bar{C},$$

where $\bar{C} = \frac{1}{n_{ta}} \sum_{i=1}^{n_{ta}} X_{ta,i} X_{ta,i}^\top$ is the empirical covariance matrix.

**Theorem 5.2.** *Assume Assumption 3.1 and $q_z = \mathcal{N}(0, \lambda^2 I_d)$. Let $\widehat{q}_z$ be the latent distribution generated by the pre-trained models with $(\widehat{V}_{ta}, \widehat{\Sigma})$ and $M = \frac{d^2 \beta^2 (d + \lambda^2)}{\lambda} \sqrt{Dd \log (Ddn_s) (d^2 \vee D)}$. Then, with probability $1 - \delta_1$, we have that for any $t \in [\delta, T]$*

$$\left\| \bar{\widetilde{V}}_{ta} \bar{\widetilde{V}}_{ta}^\top - A_{ta} A_{ta}^\top \right\|_F^2 \leq \widetilde{O} \left( \frac{d \log(\frac{1}{\delta_1})}{m_t^2 \lambda^2 + \sigma_t^2} \left( \frac{M}{d\delta \sqrt{n_s}} + \frac{d}{n_{ta}} (m_t^2 \lambda^2 + \sigma_t^2)^2 \right) \right).$$

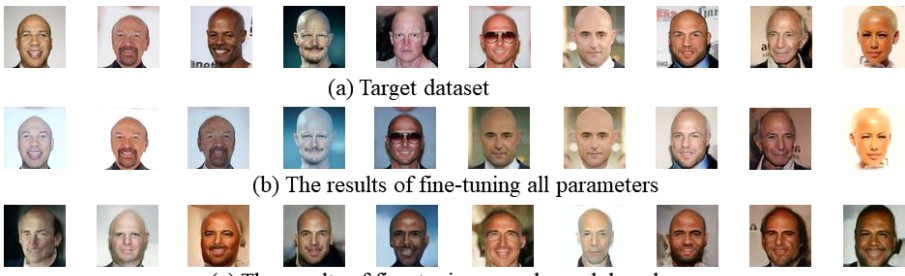

(a) Target dataset

(b) The results of fine-tuning all parameters

(c) The results of fine-tuning encoder and decoder

Figure 1: The experiments on CelebA64 dataset

The above result indicates that the few-shot diffusion models can still recover the true subspace with finite $n_s$ and $n_{ta}$. Note that when the latent distribution is Gaussian distribution, the approximation error bound for the source dataset is $n_s^{-1/2}$ instead of $n_s^{-2/d}$ (Yuan et al., 2023). Hence, $n_s$ in Theorem 5.2 do not depend on latent dimension $d$.

*Remark* 5.3. The bound of $\|VV^T - AA^T\|_F^2$ only guarantees the subspace spanned by $V$ and $A$ is close, which still holds after an orthogonal transformation on $V$. Hence, this bound does not indicate $\|V - A\|_F^2$ is small. Since all previous works (Chen et al., 2023c; Yuan et al., 2023) consider $\|VV^T - AA^T\|_F^2$, we also use this metric to measure the subspace recovery. However, our results are stronger due to the closed-form solution, where previous works do not consider how to obtain $VV^T$.

## 5.1 Discussion on the Closed-form Minimizer

**Better dependence on $n_{ta}, \delta$ and $d$.** Note that Theorem 5.2 has better $1/n_{ta}$ dependence on the target dataset compared to $1/\sqrt{n_s}$ dependence on the source dataset. Furthermore, the coefficient of $n_s$ term is dependent on the early stopping parameter $\delta$ and $D$. This is due to the $\delta$ and $D$ dependence of the approximation bound, which is used in generating $\widehat{q}_z$. However, the $n_{ta}$ term only has $d$ dependence. Hence, even in the latent Gaussian setting, we still need a larger source dataset than the target dataset to obtain a sufficiently accurate closed-form solution.

**The relationship with principal component analysis (PCA).** The expected few-shot score matching objective can be simplified to

$$\min_{V_{ta} \in \tilde{\mathcal{Q}}_{NN}(\widehat{\Sigma})} 1/\sigma_t^4 \mathbb{E}_{X_t|X_0 = X_{ta,i}} \left[ \|V_{ta}\widehat{G}_t V_{ta}^\top X_t - m_t X_0\|_2^2 \right],$$

where $\widehat{G}_t = I_d - \sigma_t^2 \widehat{\Sigma}_t^{-1}$. Note that when ignoring $1/\sigma_t^4$ and choosing $t = 0$, the above minimization problem is similar to PCA. This suggests that few-shot models implicitly optimize an objective function akin to PCA. However, few-shot models extend beyond traditional PCA. More specifically, when $\lambda^2$ is large, classical PCA suffers from the influence of $\lambda^2$. In contrast, due to $(m_t^2 \lambda^2 + \sigma_t^2)/n_{ta}$ term, few-shot models can select a large $t$ to mitigate the impact of $\lambda^2$ and achieve a $1/n_{ta}$.

## 6 Experiments

To corroborate our theoretical findings, we conducted experiments utilizing real-world datasets. These experiments show that the new model obtained by only fine-tuning appropriate encoder and decoder layers on target datasets can produce novel images with the target feature, which shows the effectiveness of the methods and supports our theoretical results.

**Datasets and benchmark.** Note that human face images tend to exhibit similarity in their latent space, primarily due to shared facial features, while differing in specific features. Hence, we initially pre-train a model using the CelebA64 dataset, focusing on distinct hairstyle features as the goal for the fine-tuning phase. For the source data, we construct a large dataset (6400 images) with different hairstyles (without the bald feature). For the target data, we choose the bald feature as the target feature and select 10 images with this feature to constitute the target dataset, which are much smaller than the size of target dataset (Figure 1 (a)). To show the effectiveness of our methods, we also fine-tune all parameters of the pre-trained models as the benchmark.

**Discussion.** As shown in Figure 1, the results obtained by only fine-tuning the encoder and decoder layers can generate novel face images with the bald feature. Conversely, when fine-tuning all parameters, the models suffer from memory phenomenon and can only generate images that slightly

modify the brightness and angle of the target dataset. This phenomenon indicates that only fine-tuning the appropriate encoder and decoder will result in a model with a generalization property.

We note that these experiments aim to verify the effectiveness of the methods instead of achieving state-of-the-art performance since previous works carefully select specific parameters, such as specific cross-attention layers (Kumari et al., 2023) or special neurons (Liu et al., 2023), to fine-tune pre-trained models. However, we simply fine-tune all encoder and decoder layers simultaneously. There are more experiments on cat faces and more discussion on why Assumption 3.1 is satisfied in our experiments. We refer to Appendix E for more details.

## 7 Conclusion

This work aims to provide a deeper understanding of few-shot diffusion models from a theoretical perspective. Our analysis is conducted from two key perspectives: the approximation and optimization aspects, all under linear structure distribution and shared latent space assumptions.

From the approximation error bound, we consider general subgaussian latent variable and prove that few-shot models have a $\widetilde{O}\left(n_s^{-2/d} + n_{ta}^{-1/2}\right)$ approximation bound, which is better than $n_{ta}^{-2/d}$ results of diffusion models without a fine-tuning phase and escape the curse of dimensionality. This result also makes the first step to explain why few-shot diffusion models only require $5 \sim 10$ images to generate great samples. The experiments on the real-world dataset also show that the fine-tuning phase only requires 10 images to generate novel images with the target feature.

When analyzing the optimization process, we consider a more special, shared Gaussian latent variable and prove that the expected score matching has a closed-form minimizer, which indicates that the few-shot diffusion models can simplify the optimization problem. Furthermore, we prove that the empirical closed-form solution has a $\widetilde{O}\left(1/n_{ta} + 1/\left(\delta\sqrt{n_s}\right)\right)$ accuracy bound, which still has better $1/n_{ta}$ target data dependence compared to $1/\left(\delta\sqrt{n_s}\right)$ dependence on the source data.

**Future work and limitation.** When considering the approximation bound, we assume a distribution with a linear structure. Though it has been supported by much empirical evidence, it is not as general as bounded distribution. After that, we plan to consider a general, bounded distribution and show the advantage of few-shot diffusion models. One possible way is to analyze the mixture of low-rank Gaussian (Wang et al., 2024), which is more general than the linear subspace assumption.

We focus on a special Gaussian latent distribution when considering the optimization problem. As a next step, we plan to consider a more general latent distribution, such as a log-concave distribution. In this setting, we can not directly obtain the closed-form solution. However, due to the shared information and simplified landscape, it is still possible to use some optimization algorithms, such as gradient descent, to optimize the few-shot objective function to achieve the convergence guarantee.

**Broader Impact.** This paper presents work whose goal is to understand few-shot diffusion models from the theoretical perspective. A noteworthy societal impact is that few-shot diffusion models may be used to imitate the style of artists and generate fake images, thereby infringing on the rights of artists (Mirsky & Lee, 2021). We recommend adding watermarks to images to determine whether the image was generated by a generative model (Fernandez et al., 2023). The other societal impact is the same as general generative models (Mishkin et al., 2022).

## Acknowledgments and Disclosure of Funding

The author Bo Jiang is supported by National Natural Science Foundation of China (62072302). The author Baoxiang Wang is partially supported by the National Natural Science Foundation of China (62106213, 72394361), an extended support project from the Shenzhen Science and Technology Program, and support from Longgang District Key Laboratory of Intelligent Digital Economy Security.

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

# Appendix

## A  The Neural Network Structure

In this section, we introduce the multi-layer ReLU network $\mathbf{f}_{\boldsymbol{\theta}} \in \mathrm{NN}\left(L, M', J, K_1, \kappa, \gamma, \gamma_t\right)$ in $\mathcal{S}_{\mathrm{NN}}$. We note that the following setting is exactly the same as the one in Chen et al. (2023c), and we show the structure for completeness. We denote by $\mathrm{NN}\left(L, M', J, K_1, \kappa, \gamma, \gamma_t\right)$ the following neural network:

$$\mathrm{NN}\left(L, M', J, K_1, \kappa, \gamma, \gamma_t\right) = \{\mathbf{f}(Z, t) = W_L \sigma\left(\ldots \sigma\left(W_1\left[Z^\top, t\right]^\top + \mathbf{b}_1\right)\ldots\right) + \mathbf{b}_L :$$

$$\text{network width bounded by } M', \sup_{z,t}\|\mathbf{f}(z,t)\|_2 \le K_1,$$

$$\max\left\{\|\mathbf{b}_i\|_\infty, \|W_i\|_\infty\right\} \le \kappa \text{ for } i = 1, \ldots, L,$$

$$\sum_{i=1}^{L}\left(\|W_i\|_0 + \|\mathbf{b}_i\|_0\right) \le J,$$

$$\|\mathbf{f}\left(Z_1, t\right) - \mathbf{f}\left(Z_2, t\right)\|_2 \le \gamma\|Z_1 - Z_2\|_2 \text{ for any } t \in [0, T],$$

$$\|\mathbf{f}\left(Z, t_1\right) - \mathbf{f}\left(Z, t_2\right)\|_2 \le \gamma_t\left|t_1 - t_2\right| \text{ for any } Z\},$$

where $\sigma$ is the ReLU activation function. Given an network accuracy $\epsilon > 0$, the parameters is defined by

$$L = \mathcal{O}\left(\log\frac{1}{\epsilon} + d\right), K_1 = \mathcal{O}\left(2d^2\log\left(\frac{d}{\delta\epsilon}\right)\right),$$

$$M' = \mathcal{O}\left((1+\beta)^d T\tau d^{d/2+1}\epsilon^{-(d+1)}\log^{d/2}\left(\frac{d}{\delta\epsilon}\right)\right),$$

$$J = \mathcal{O}\left((1+\beta)^d T\tau d^{d/2+1}\epsilon^{-(d+1)}\log^{d/2}\left(\frac{d}{\delta\epsilon}\right)\left(\log\frac{1}{\epsilon} + d\right)\right),$$

$$\kappa = \mathcal{O}\left(\max\left\{2(1+\beta)\sqrt{d\log\left(\frac{d}{\delta\epsilon}\right)}, T\tau\right\}\right),$$

$$\gamma = 10d(1+\beta), \gamma_t = 10\tau,$$

where $\tau = \sup_{A \in \{A_s, A_{ta}\}} \sup_{t \in [\delta, T]} \sup_{\|z\|_\infty \le \sqrt{d\log\frac{d}{\delta\epsilon}}} \left\|\frac{\partial}{\partial t}\left[\sigma_t A\nabla\log q_t^{\mathrm{LD}}\left(z\right)\right]\right\|_2$.

## B  The Proof of the Approximation Error Bound

Let $(\widehat{V}_s, \widehat{\theta})$ be the minimizer of the pre-trained objective function. The few-shot diffusion model freezes the bottleneck network and fine-tunes $V_{ta} \in \mathbb{R}^{D \times d}$ to obtain the minimizer $(\widehat{V}_{ta}, \widehat{\theta})$ of the few-shot objective function. As the first step, we show that with the bottleneck parameterized by $\widehat{\theta}$, there also exists a solution $(\bar{V}_{ta}, \widehat{\theta}) \in \mathcal{Q}_{NN}(\widehat{\theta})$ achieve the $\epsilon^2 + n_s^{-2/d}$ error bound.

**Lemma B.1.** *If* $\epsilon \le n_s^{-\frac{1-\alpha(n_s)}{d+5}}$, *where* $\alpha(n) = \frac{d\log\log n}{\log n}$, *then with probability* $1 - \delta_1$, *there exists a solution* $(\bar{V}_{ta}, \widehat{\theta}) \in \mathcal{Q}_{NN}(\widehat{\theta})$ *such that*

$$\int_\delta^T \mathbb{E}_{X \sim q_t^{ta}}\left[\left\|\mathbf{s}_{\bar{V}_{ta}, \widehat{\theta}}(X, t) - \nabla\log q_t^{ta}(X)\right\|_2^2\right]\mathrm{d}t \le O\left(\frac{d}{\delta}\epsilon^2 + \frac{(T-\delta)(d + C_Z)d^2\beta^2}{\delta^2 c_0}n_s^{-\frac{2-2\alpha(n_s)}{d+5}}\log(\frac{1}{\delta_1})\right).$$

**Proof.** As shown in Theorem 1 of Chen et al. (2023c), there exists a solution $(\bar{V}_s, \bar{\theta})$ in $\mathcal{S}_{NN}$ such that

$$\left\|\mathbf{s}_{\bar{V}_s, \bar{\theta}}(\cdot, t) - \nabla\log q_t^s(\cdot)\right\|_2 \le \frac{\sqrt{d}+1}{\sigma_t^2}\epsilon, \forall t \in [\delta, T].$$

Hence, we do the following decomposition:

$$\left\| \mathbf{s}_{\bar{V}_{ta},\hat{\theta}}(\cdot, t) - \nabla \log q_t^{ta}(\cdot) \right\|_2^2 \lesssim \left\| \mathbf{s}_{\bar{V}_{ta},\bar{\theta}}(\cdot, t) - \nabla \log q_t^{ta}(\cdot) \right\|_2^2 + \left\| \mathbf{s}_{\bar{V}_{ta},\hat{\theta}}(\cdot, t) - \mathbf{s}_{\bar{V}_{ta},\bar{\theta}}(\cdot, t) \right\|_2^2$$

For the encoder and decoder layer, we choose $\bar{V}_{ta} = A_{ta}$. The first term is bounded due to the construction of the neural network. We first show that $\mathbf{s}_{\bar{V}_{ta},\bar{\theta}}$ is $\epsilon$-close to the true score function $\nabla \log q_t^{ta}$. Since the encoder and decoder have been chosen, we only need to focus on the latent bottleneck. For the latent bottleneck, we need to use $\mathbf{f}_\theta(Z, t)$ to approximate ground-truth function $h(Z, t) = \sigma_t^2 \nabla \log q_t^{\mathrm{LD}}(Z) + Z$ for $Z \in \mathbb{R}^d$. Chen et al. (2023c) show that for any latent variable $Z' \in \mathbb{R}^d$ with subgaussian tail, we have that

$$\| h(Z', t) - \mathbf{f}_{\bar{\theta}}(Z', t) \|_{L^2(q_t^{\mathrm{LD}})} \le (\sqrt{d} + 1)\epsilon .$$

Then, we have that

$$\left\| \mathbf{s}_{\bar{V}_{ta},\bar{\theta}}(\cdot, t) - \nabla \log q_t^{ta}(\cdot) \right\|_2^2 \le \frac{d}{\sigma_t^4} \epsilon^2 .$$

For the second term, we know that with probability $1 - \delta_1$:

$$\int_\delta^T \mathbb{E}_{X \sim q_t} \left[ \left\| \mathbf{s}_{\bar{V}_{ta},\bar{\theta}}(\cdot, t) - \mathbf{s}_{\bar{V}_{ta},\hat{\theta}}(\cdot, t) \right\|_2^2 \right]$$

$$\le \int_\delta^T \frac{1}{\sigma_t^4} \mathbb{E}_{Z \sim q_t^{\mathrm{LD}}} \left[ \| \mathbf{f}_{\hat{\theta}}(Z) - \mathbf{f}_{\bar{\theta}}(Z) \|_2^2 \right] \mathrm{d}t$$

$$\le O \left( \frac{T - \delta}{\delta^2} \left( \frac{\delta}{c_0} \left( (T - \log \delta) d \cdot \gamma^2 + d\beta \right) + \frac{\gamma^2 \cdot C_Z}{c_0} \right) n_s^{-\frac{2 - 2\alpha(n_s)}{d+5}} \right) \log \left( \frac{1}{\delta_1} \right) ,$$

where $\alpha(n) = \frac{d \log \log n}{\log n}$. The first inequality follows $A_{ta}$ is a matrix with orthonormal columns. Since we assume $\epsilon \le n_s^{-\frac{1 - \alpha(n_s)}{d+5}}$, the network has good enough ability to obtain an accurate enough $\hat{\theta}$. Hence, we can use Appendix C.4 of Chen et al. (2023c) to obtain the second inequality. Since we directly use the true matrix $\bar{V}_{ta} = A_{ta}$ instead of the approximate $\hat{V}_{ta}$, we do not need orthogonal transformation and can choose $U = I_d$ in the Appendix C.4 of Chen et al. (2023c). Then, we complete our proof. ∎

To prove Theorem 4.3, we need to do the following decomposition for the population loss of the target datasets

$$\mathcal{L}_{ta} \left( \mathbf{s}_{\hat{V}_{ta},\hat{\theta}} \right)$$

$$= \mathcal{L}_{ta} \left( \mathbf{s}_{\hat{V}_{ta},\hat{\theta}} \right) - (1 + a)\hat{\mathcal{L}}_{ta} \left( \mathbf{s}_{\hat{V}_{ta},\hat{\theta}} \right) + (1 + a)\hat{\mathcal{L}}_{ta} \left( \mathbf{s}_{\hat{V}_{ta},\hat{\theta}} \right)$$

$$\le \underbrace{\mathcal{L}_{ta}^{\mathrm{trunc}} \left( \mathbf{s}_{\hat{V}_{ta},\hat{\theta}} \right) - (1 + a)\hat{\mathcal{L}}_{ta}^{\mathrm{trunc}} \left( \mathbf{s}_{\hat{V}_{ta},\hat{\theta}} \right)}_{(a)} + \underbrace{\mathcal{L}_{ta} \left( \mathbf{s}_{\hat{V}_{ta},\hat{\theta}} \right) - \mathcal{L}_{ta}^{\mathrm{trunc}} \left( \mathbf{s}_{\hat{V}_{ta},\hat{\theta}} \right)}_{(b)} + (1 + a) \underbrace{\inf_{s_{V_{ta},\hat{\theta}} \in \mathcal{Q}(\hat{\theta})} \hat{\mathcal{L}}_{ta} \left( \mathbf{s}_{V_{ta},\hat{\theta}} \right)}_{(c)} ,$$

where $a \in (0, 1)$ and $\mathcal{L}_{ta}^{\mathrm{trunc}}$ is defined as

$$\mathcal{L}_{ta}^{\mathrm{trunc}} \left( \mathbf{s}_{\hat{V}_{ta},\hat{\theta}} \right) = \mathbb{E}_{x \sim q_0} \left[ \ell_{ta}^{\mathrm{trunc}} \left( x; \mathbf{s}_{\hat{V}_{ta},\hat{\theta}} \right) \right] = \mathbb{E}_{x \sim q_0} \left[ \ell_{ta} \left( x; \mathbf{s}_{\hat{V}_{ta},\hat{\theta}} \right) \mathbf{1} \left\{ \|x\|_2 \le R \right\} \mathrm{d}t \right] .$$

In this section, we take $R = \mathcal{O} \left( \sqrt{d \log d + \log K_1 + \log \frac{n_{ta}}{\delta_1}} \right)$ to guarantee $\mathbb{P}_{X_{ta,i} \sim q^{ta}} \left( \|X_{ta,i}\|_2 \le R \text{ for all } i = 1, \ldots, n_{ta} \right) \ge 1 - \delta_1$, where $K_1$ is defined in Appendix A.

**Term (a).** Similar to Chen et al. (2023c), we define a function class $\mathcal{G}(\hat{\theta}) = \left\{ \ell_{ta}^{\mathrm{trunc}} \left( \cdot; \mathbf{s}_{V,\hat{\theta}} \right) : \mathbf{s}_{V,\hat{\theta}} \in \mathcal{Q}_{\mathrm{NN}}(\hat{\theta}) \right\}$, which is induced by $\mathcal{Q}(\hat{\theta})$. For the upper bound of $\mathcal{G}(\hat{\theta})$, we directly use the augmentation of Chen et al. (2023c) to obtain that

$$\ell_{ta}^{\mathrm{trunc}} \left( X; \mathbf{s}_{V,\hat{\theta}} \right) \le \mathcal{O} \left( \frac{K_1^2 + R^2}{\delta (T - \delta)} \right) , \text{ for any } \mathbf{s}_{V,\hat{\theta}} \in \mathcal{Q}_{\mathrm{NN}}(\hat{\theta}) .$$

Then, by using Lemma D.1, we know that with probability $1 - \delta_1$, term (a) is bounded by

$$
\mathcal{O}\left( \frac{(1 + 3/a)\left((1+\beta)^2 d^2 \log \frac{d}{\delta \epsilon} + \log \frac{n_{ta}}{\delta}\right)}{n_{ta}\delta\,(T-\delta)} \log \frac{\mathcal{N}\left(\tau_1, \mathcal{G}(\hat{\theta}), \|\cdot\|_\infty\right)}{\delta_1} + \tau_1 \right).
$$

To bound the above term, we need to calculate the covering number of $\mathcal{G}(\hat{\theta})$, which is related to a $\iota$-covering of $\mathcal{Q}_{NN}(\hat{\theta})$. Suppose that given $\mathbf{s}_{V_1,\hat{\theta}}$ and $\mathbf{s}_{V_2,\hat{\theta}}$ with $\sup_{\|x\|_2 \le 3R + \sqrt{D\log D}, t \in [\delta, T]} \left\| \mathbf{s}_{V_1,\hat{\theta}}(x,t) - \mathbf{s}_{V_2,\hat{\theta}}(x,t) \right\|_2 \le \iota$, we need to bound

$$
\left\| \ell_{ta}^{\mathrm{trunc}}\left(\cdot; \mathbf{s}_{V_1,\hat{\theta}}\right) - \ell_{ta}^{\mathrm{trunc}}\left(\cdot; \mathbf{s}_{V_2,\hat{\theta}}\right) \right\|_\infty.
$$

By using the same calculation compared to Term (A) of Chen et al. (2023c), we know that

$$
\left\| \ell_{ta}^{\mathrm{trunc}}\left(\cdot; \mathbf{s}_{V_1,\hat{\theta}}\right) - \ell_{ta}^{\mathrm{trunc}}\left(\cdot; \mathbf{s}_{V_2,\hat{\theta}}\right) \right\|_\infty \le \mathcal{O}\left( \frac{\iota}{T-\delta}(K_1 + R)\log\frac{T}{\delta} + \frac{4K_1(K_1+R)}{\delta\,(T-\delta)}(R/D)^{D-2}\exp\left(-\frac{1}{\sigma_t^2}R^2\right) \right).
$$

The above inequality indicates that a $\iota$-covering of $\mathcal{Q}_{NN}(\hat{\theta})$ in $L_2$ norm leads a $\frac{\iota}{T-\delta}(K_1+R)\log\frac{T}{\delta} + \frac{4K_1(K_1+R)}{\delta(T-\delta)}(R/D)^{D-2}\exp\left(-\frac{1}{\sigma_t^2}R^2\right)$-covering of of $\mathcal{G}(\hat{\theta})$ in $L_\infty$ norm.

By taking $R = \mathcal{O}\left(\sqrt{d\log d + \log K_1 + \log\frac{n_{ta}}{\delta_1}}\right)$, $K_1 = \mathcal{O}\left(2d^2 \log\left(\frac{d}{\delta\epsilon}\right)\right)$, $\iota = \frac{2}{n_{ta}\delta(T-\delta)}$, we know that

$$
\tau_1 \le \frac{d^2}{n_{ta}\delta} \log(\frac{T}{\delta}) \log(\frac{d}{\delta\epsilon}) \log(\frac{n_{ta}}{\delta}),
$$

which indicates with probability $1 - \delta_1$, term (a) is bounded by

$$
\mathcal{O}\left( \frac{(1+3/a)\left((1+\beta)^2 d^2 \log\frac{d}{\delta\epsilon} + \log\frac{n_{ta}}{\delta}\right)}{n_{ta}\delta\,(T-\delta)} \log\frac{\mathcal{N}\left(\frac{1}{n_{ta}\delta(T-\delta)}, \mathcal{Q}_{NN}(\hat{\theta}), \|\cdot\|_2\right)}{\delta_1} + \frac{d^2}{n_{ta}\delta}\log(\frac{T}{\delta})\log(\frac{d}{\delta\epsilon})\log(\frac{n_{ta}}{\delta}) \right).
$$

After that, we need to determine the covering number of $\mathcal{Q}_{NN}(\hat{\theta})$ with a truncated $X$ to bound term (a).

**Lemma B.2.** *The logarithmic covering number of $\mathcal{Q}_{NN}(\theta)$ for $\|X\|_2 \le 3R + \sqrt{D\log D}, t \in [\delta, T]$ is*

$$
\log \mathcal{N}\left(\iota, \mathcal{Q}_{NN}(\theta), \|\cdot\|_2\right) = \mathcal{O}\left( 2Dd \cdot \log\left( 1 + \frac{6K\gamma\sqrt{d}(3R + \sqrt{D\log D})}{\delta\iota} \right) \right).
$$

**Proof.** Suppose that there exists two orthonormal column matrix $V_1, V_2$ such that $\|V_1 - V_2\|_F \le \delta_2$, then we have

$$
\sup_{\|X\|_2 \le 3R + \sqrt{D\log D}, t \in [\delta, T]} \|\mathbf{s}_{V_1,\theta}(X,t) - \mathbf{s}_{V_2,\theta}(X,t)\|_2
$$

$$
= \frac{1}{\sigma_t^2} \sup_{\|X\|_2 \le 3R + \sqrt{D\log D}, t \in [\delta, T]} \left[\left\| V_1\mathbf{f}_\theta\left(V_1^\top X, t\right) - V_1\mathbf{f}_\theta\left(V_2^\top X, t\right) \right\|_2 + \left\| V_1\mathbf{f}_\theta\left(V_2^\top x, t\right) - V_2\mathbf{f}_\theta\left(V_2^\top X, t\right) \right\|_2\right]
$$

$$
\le \frac{1}{\sigma_t^2}\left( \gamma\delta_1\sqrt{d}(3R + \sqrt{D\log D}) + \delta_1 K \right)
$$

For set $\left\{ V \in \mathbb{R}^{D \times d} : \|V\|_2 \le 1 \right\}$, the $\delta_2$-covering number is $\left( 1 + 2\frac{\sqrt{d}}{\delta_2} \right)^{Dd}$. Then we know that

$$
\log \mathcal{N}\left(\iota, \mathcal{S}_{NN}, \|\cdot\|_2\right) = \mathcal{O}\left( 2Dd \cdot \log\left( 1 + \frac{6K\gamma\sqrt{d}(3R + \sqrt{D\log D})}{\delta\iota} \right) \right).
$$

∎

**Term (b).** For the term (b), the proof of Theorem 2 in Chen et al. (2023c) shows that

$$\text{Term (b)} \le \frac{1}{n_{ta}\delta\,(T-\delta)}\,.$$

**Term (c).** For the term (c), we know that it is bounded by the constructed solution $(\bar{V}_{ta}, \hat{\theta})$:

$$\inf_{s_{V_{ta},\hat{\delta}}\in\mathcal{Q}(\hat{\theta})} \widehat{\mathcal{L}}_{ta}\left(\mathbf{s}_{V_{ta},\hat{\theta}}\right) \le \underbrace{\widehat{\mathcal{L}}_{ta}\left(\mathbf{s}_{\bar{V}_{ta},\hat{\theta}}\right) - (1+a)\mathcal{L}^{\text{trunc}}\left(\mathbf{s}_{\bar{V}_{ta},\hat{\theta}}\right)}_{(C_1)} + (1+a)\underbrace{\mathcal{L}^{\text{trunc}}\left(\mathbf{s}_{\bar{V}_{ta},\hat{\theta}}\right)}_{(C_2)}\,.$$

For the term (C.1), since $\mathbf{s}_{\bar{V}_{ta},\hat{\theta}}$ is a fixed function, we directly use the results of (Chen et al., 2023c):

$$\text{Term}(C_1) \le O\left(\frac{(1+6/a)\left((1+\beta)^2 d^2 \log\frac{d}{\delta\epsilon} + \log\frac{n}{\delta}\right)}{n_{ta}\delta\,(T-\delta)}\log\frac{1}{\delta_1}\right)\,,$$

with probability $1-\delta_1$. For the term (C.2), we know that

$$\mathcal{L}^{\text{trunc}}_{ta}\left(\mathbf{s}_{\bar{V}_{ta},\hat{\theta}}\right) \le \mathcal{L}\left(\mathbf{s}_{\bar{V}_{ta},\hat{\theta}}\right) = \frac{1}{T-\delta}\int_\delta^T \left\|\mathbf{s}_{\bar{V}_{ta},\hat{\theta}}(\cdot,t) - \nabla\log q_t^{ta}(\cdot)\right\|^2_{L^2(q_t)} dt$$

$$+ \underbrace{\mathcal{L}\left(\mathbf{s}_{\bar{V}_{ta},\hat{\theta}}\right) - \frac{1}{T-\delta}\int_\delta^T \left\|\mathbf{s}_{\bar{V}_{ta},\hat{\theta}}(\cdot,t) - \nabla\log q_t^{ta}(\cdot)\right\|^2_{L^2(q_t)} dt}_{(\mathcal{E})}\,.$$

As we show in Section 3.2, the two terms in $\mathcal{E}$ are both the score matching objective function and have a constant different $E$, which is independent of the trainable parameters $(V,\theta)$. We denote by this difference $E$ and $F = \frac{(d+C_Z)d^2\beta^2}{\delta^2 c_0}$. With probability $1-\delta_1$, Lemma B.1 shows that term (C.2) is bounded by

$$O\left(\frac{d}{\delta(T-\delta)}\epsilon^2 + F n_s^{-\frac{2-2\alpha(n_s)}{d+5}}\log\left(\frac{1}{\delta_1}\right)\right) + E\,.$$

After bounding these three terms, we prove Theorem 4.3.

**Theorem 4.3.** *Let* $\alpha(n) = \frac{d\log\log n}{\log n}$, $F = \frac{(d+C_Z)d^2\beta^2}{\delta^2 c_0}$ *and network parameter* $\epsilon = n_{ta}^{-1/2}$. *Assume Assumption 3.1, 4.1, 4.2 and* $n_{ta}^{\frac{d+5}{4(1-\alpha(n_s))}} \ge n_s$. *Then, with probability* $1-\delta_1$, *the following inequality holds (hiding logarithmic factors)*

$$\frac{1}{T-\delta}\int_\delta^T \left\|\mathbf{s}_{\hat{V}_{ta},\hat{\theta}}(\cdot,t) - \nabla\log q_t^{ta}(\cdot)\right\|^2_{L^2(q_t^{ta})} dt \le \tilde{O}\left(\left(\frac{(1+\beta)^2 D d^3}{\delta\,(T-\delta)\sqrt{n_{ta}}} + F n_s^{-\frac{2-2\alpha(n_s)}{d+5}}\right)\log\left(\frac{1}{\delta_1}\right)\right)\,.$$

**Proof.** Equipped with the bound of the term (a), (b), and (c) and hiding the logarithmic term (except for the covering number term), with probability $1-\delta_1$, we have that

$$\mathcal{L}_{ta}\left(\mathbf{s}_{\hat{V}_{ta},\hat{\theta}}\right) \le (1+a)^2 E + \tilde{O}\left(\frac{\left((1+\beta)^2 d^2 \log\frac{d}{\delta\epsilon} + \log\frac{n_{ta}}{\delta}\right)}{a\delta\,(T-\delta)\,n_{ta}}\log\frac{\mathcal{N}\left(\frac{1}{n_{ta}\delta(T-\delta)},\mathcal{Q}_{\text{NN}}(\hat{\theta}),\|\cdot\|_2\right)}{\delta_1} + \frac{d^2}{n_{ta}\delta}\right.$$

$$\left. + \frac{1}{n_{ta}\delta(T-\delta)} + \frac{d}{\delta(T-\delta)}\epsilon^2 + F n_s^{-\frac{2-2\alpha(n_s)}{d+5}}\log\left(\frac{1}{\delta_1}\right)\right)\,.$$

Since $\mathcal{L}_{ta}\left(\mathbf{s}_{\widehat{V}_{ta},\widehat{\theta}}\right) - E = \frac{1}{T-\delta}\int_\delta^T \left\|\mathbf{s}_{\widehat{V}_{ta},\widehat{\theta}}(\cdot,t) - \nabla\log q_t^{ta}(\cdot)\right\|_{L^2(q_t)}^2$, we have that the following inequality when choosing $a = \epsilon^2$:

$$\frac{1}{T-\delta}\int_\delta^T \left\|\mathbf{s}_{\widehat{V}_{ta},\widehat{\theta}}(\cdot,t) - \nabla\log q_t^{ta}(\cdot)\right\|_{L^2(q_t)}^2$$

$$\leq \tilde{O}\left(\frac{(1+\beta)^2 d^2}{\epsilon^2\delta\,(T-\delta)\,n_{ta}}\log\frac{\mathcal{N}\left(\frac{1}{n_{ta}\delta(T-\delta)},\mathcal{Q}_{\mathrm{NN}}(\widehat{\theta}),\|\cdot\|_2\right)}{\delta_1} + \frac{d}{\delta(T-\delta)}\epsilon^2 + \frac{d^2}{n_{ta}\delta} + Fn_s^{-\frac{2-2\alpha(n_s)}{d+5}}\log\left(\frac{1}{\delta_1}\right)\right)$$

$$\leq \tilde{O}\left(\frac{(1+\beta)^2 Dd^3}{\delta\,(T-\delta)\,n_{ta}\epsilon^2}\log\left(1+\frac{6K\gamma\sqrt{d}(3R+\sqrt{D\log D})n_{ta}\delta(T-\delta)}{\delta_1}\right) + \frac{d}{\delta(T-\delta)}\epsilon^2 + \frac{d^2}{n_{ta}\delta}\right.$$

$$\left. + Fn_s^{-\frac{2-2\alpha(n_s)}{d+5}}\log\left(\frac{1}{\delta_1}\right)\right),$$

where the second inequality follows the convering number of $\mathcal{Q}_{NN}(\widehat{\theta})$ for $\|X\| \leq 3R + \sqrt{D\log D}$ with $R = \mathcal{O}\left(\sqrt{d\log d + \log K_1 + \log\frac{n_{ta}}{\delta_1}}\right)$ and the network parameters is defined in Appendix A. Finally, we choose $\epsilon^2 = 1/\sqrt{n_{ta}}$, then we have that

$$\frac{1}{T-\delta}\int_\delta^T \left\|\mathbf{s}_{\widehat{V}_{ta},\widehat{\theta}}(\cdot,t) - \nabla\log q_t^{ta}(\cdot)\right\|_{L^2(q_t)}^2 \leq \tilde{O}\left(\frac{(1+\beta)^2 Dd^3}{\delta\,(T-\delta)\sqrt{n_{ta}}}\log\left(\frac{1}{\delta_1}\right) + \frac{d^2}{n_{ta}\delta} + Fn_s^{-\frac{2-2\alpha(n_s)}{d+5}}\log\left(\frac{1}{\delta_1}\right)\right).$$

As we require in Lemma B.1, we need $\epsilon = 1/n_{ta}^{1/4} \leq n_s^{-\frac{1-\alpha(n_s)}{d+5}}$, which indicates $n_{ta}^{\frac{d+5}{4(1-\alpha(n_s))}} \geq n_s$. ∎

## C  The Proof of the Optimization Problem

### C.1  The Pre-trained Diffusion Model Generate Accurate Enough Latent Distribution

Since we need to use the approximated latent distribution in the few-shot phase, we show that the pre-trained diffusion models with solution $(\widehat{V}_s, \widehat{\theta})$ can generate an accurate enough latent distribution. As shown in Section 5, when considering the optimization perspective of diffusion models, we assume the latent distribution is a Gaussian distribution $q_z = \mathcal{N}(0, \Sigma)$ with $\Sigma = \mathrm{diag}\left(\lambda_1^2,\ldots,\lambda_d^2\right) \succ 0$. Yuan et al. (2023) show that in the setting, the approximation error bound (Lemma D.3) for the target dataset is

$$\frac{1}{T-\delta}\int_\delta^T \left\|\nabla\log q_t^s(\cdot) - \mathbf{s}_{\widehat{V}_s,\widehat{\theta}}(\cdot,t)\right\|_{L^2(q_t^s)}^2 \mathrm{d}t \leq O\left(\frac{1}{\delta}\sqrt{\frac{(d^2+Dd)\log(Ddn_s)(d^2 \vee D)\log\frac{1}{\delta_1}}{n_s}}\right).$$

To generate latent distribution, we first introduce the reverse process in the latent space. The introduction mainly follows the outline of Appendix C.2 of Chen et al. (2023c). For $X_t$, we can do the following decomposition: $X_t = A_s Z_t + X_{t,\perp}$, where $Z_t = A_s^\top X_t$. With $Z_t^\leftarrow = Z_{T-t}$, the reverse process in the latent space is

$$\mathrm{d}Z_t^\leftarrow = \left[\frac{1}{2}Z_t^\leftarrow + \nabla\log q_{T-t}^{\mathrm{LD}}\left(Z_t^\leftarrow\right)\right]\mathrm{d}t + \mathrm{d}\left(A_s^\top B_t\right)$$

As shown in Theorem 3 of Chen et al. (2023c), the solution $(\widehat{V}_s, \widehat{\theta})$ of the pre-trained diffusion models only guarantee $\|\widehat{V}_s\widehat{V}_s^\top - A_s A_s^\top\|_F^2$ is small instead of $\|\widehat{V}_s - A_s\|_F^2$ is small. Hence, Theorem 3 of Chen et al. (2023c) assume there exists an orthogonal matrix $U_s \in \mathbb{R}^{d\times d}$ and do an orthogonal transformation on $\widehat{V}_s$ to obtain $\widehat{V}_s U_s$, which can guarantee $\|\widehat{V}_s U_s - A_s\|_F^2$ is small. After such orthogonal transformation, the reverse process with an approximated score function and an approximated reversing beginning distribution $\widetilde{Z}_0^{\leftarrow,r} \sim \mathcal{N}(0, I_d)$ is

$$\mathrm{d}\widetilde{Z}_t^{\leftarrow,r} = \left[\frac{1}{2}\widetilde{Z}_t^{\leftarrow,r} + \mathbf{s}_{U_s,\widehat{\theta}}^{\mathrm{LD}}\left(\widetilde{Z}_t^{\leftarrow,r}, T-t\right)\right]\mathrm{d}t + \mathrm{d}\left(U_s^\top\widehat{V}_s^\top B_t\right), \quad \widetilde{Z}_0^{\leftarrow,r} \sim \mathrm{N}(0, I_d) \qquad (2)$$

where

$$\widetilde{Z}_t^{\leftarrow,r} = U_s^\top \widetilde{Z}_t^\leftarrow \text{ and } \mathbf{s}_{U_s,\widehat{\theta}}^{\mathrm{LD}}(Z,t) = \frac{1}{\sigma_t^2}\left[-Z + U_s^\top \mathbf{f}_{\widehat{\theta}}(U_s Z, t)\right].$$

Then, we discretize the above process with the exponential integrator (EI) discretization scheme (Zhang & Chen, 2022) to obtain an implementable algorithm:

$$\mathrm{d}\widetilde{Z}_t^{\leftarrow,r} = \left[\frac{1}{2}\widetilde{Z}_t^{\leftarrow,r} + \mathbf{s}_{U_s,\widehat{\theta}}^{\mathrm{LD}}\left(\widetilde{Z}_{k\eta}^{\leftarrow,r}, T - t_k'\right)\right]\mathrm{d}t + \mathrm{d}\left(U_s^\top \widehat{V}_s^\top B_t\right), \text{ where } t \in [t_k', t_{k+1}']. \quad (3)$$

As shown in Appendix C.4 of Chen et al. (2023c), if the target ground truth score function has a $L_2$-accurate approximated score:

$$\frac{1}{T-\delta}\int_\delta^T \left\|\mathbf{s}_{\widehat{V}_s,\widehat{\theta}}(\cdot,t) - \nabla\log q_t^s(\cdot)\right\|_{L^2(q_t^s)}^2 \mathrm{d}t \leq \epsilon^2,$$

the latent score function also has an $L_2$ norm bound $\epsilon_{\text{latent-score}}^2$, which is determined by $\epsilon$:

$$\epsilon_{\text{latent-score}} = \epsilon \cdot \mathcal{O}\left(\left[\frac{\delta}{c_0}\left((T - \log\delta)d\cdot\gamma^2 + d\beta\right) + \frac{\gamma^2 \cdot C_Z}{c_0}\right]\right).$$

The remaining term is to determined $\epsilon$. Since we assume Gaussian latent variable instead of sub-Gaussian one. Hence, we do not use $\epsilon$ in Chen et al. (2023c). We use $\epsilon$ in Yuan et al. (2023) (Theorem 4.5 of Yuan et al. (2023)), which also considers Gaussian latent variable, to achieve the final results.

Finally, we have that with probability $1 - \delta_1$:

$$\frac{1}{T-\delta}\int_\delta^T \left\|\nabla\log q_t^{\mathrm{LD}}(\cdot) - \mathbf{s}_{U_s,\widehat{\theta}}^{\mathrm{LD}}(\cdot,t)\right\|_{L^2(q_t^{\mathrm{LD}})}^2 \mathrm{d}t$$
$$\leq O\left(\frac{d^2\beta^2\left(d + \lambda_{\max}^2\right)}{\lambda_{\min}\delta}\sqrt{\frac{(d^2 + Dd)\log(Ddn_s)(d^2\vee D)\log\frac{1}{\delta_1}}{n_s}}\right) \triangleq \epsilon_{\text{latent-score}}^2. \quad (4)$$

Let $p_t^{\mathrm{LD}}$ be the distribution of the algorithm (the above discretization process). In the following lemma, we adopt Theorem 5 of Chen et al. (2023a) and show that the pre-trained diffusion model can obtain an accurate enough latent distribution with the above $L_2$-accurate latent score function.

**Lemma C.1.** *With $\epsilon_{\text{latent-score}}^2$ defined in Equation (4), $T = \log\left(\frac{\lambda_{max}^2 + d}{\epsilon_{\text{latent-score}}^2}\right)$ and $K = \Theta\left(\frac{d^2(T + \log(\lambda_{max}^2))^2}{\epsilon_{\text{latent-score}}^2}\right)$, by using the exponentially decreasing (then the constant) stepsize $h_k = c\min\left\{\max\left\{t_k, \frac{1}{\lambda_{max}^2}\right\}, 1\right\}, c = \frac{\log(\lambda_{max}^2) + T}{K}$, the results $p_T^{\mathrm{LD}}$ of sampling algorithm (Equation (3)) has the following guarantee with probability $1 - \delta_1$ (hiding the logarithmic factor):*

$$\mathrm{KL}\left(q_0\|\hat{p}_T^{\mathrm{LD}}\right) \leq \widetilde{O}(\epsilon_{\text{latent-score}}^2)$$
$$= \widetilde{O}\left(\frac{d^2\beta^2\left(d + \lambda_{max}^2\right)}{\lambda_{min}\delta}\sqrt{\frac{(d^2 + Dd)\log(Ddn_s)(d^2\vee D)\log\frac{1}{\delta_1}}{n_s}}\right)$$

**Proof.** The Theorem 5 of Chen et al. (2023a) show that if $\nabla\log q_0^{\mathrm{LD}}$ is $L$-Lipschitz, diffusion models with a $L_2$-accurate can generate $\hat{p}_T^{\mathrm{LD}}$, which is close to $q_0$ in KL divergence. Since $q_z = \mathcal{N}(0,\Sigma)$, it is easy to verify $L = \lambda_{\max}^2$. Then, we complete the proof. ∎

### C.2 The Closed-form Minimizer of Few-shot Diffusion Models

When the latent distribution is a Gaussian distribution $q_z = \mathcal{N}(0,\Sigma)$ with $\Sigma = \mathrm{diag}\left(\lambda_1^2,\ldots,\lambda_d^2\right) \succ 0$, the ground truth score function for the target dataset is

$$\nabla\log q_t^{ta}(X) = -A_{ta}\Sigma_t^{-1}A_{ta}^\top X - \frac{1}{\sigma_t^2}\left(I_D - A_{ta}A_{ta}^\top\right)X,$$

where $\Sigma_t = \mathrm{diag}\left(\ldots, m_t^2\lambda_k^2 + \sigma_t^2, \ldots\right)$. Since the matrix $A_{ta}$ is independent of time $t$, we fix a $t \in [\delta, T]$ and minimize the few-shot objective function at this time. With an approximated $\widehat{\Sigma}$, which is learned by the pre-trained diffusion models, $\mathbf{f}_{\hat{\theta}}(Z, t) = (I_d - \sigma_t^2\widehat{\Sigma}_t^{-1})Z$, where $\widehat{\Sigma}_t = m_t^2\widehat{\Sigma} + \sigma_t^2 I_d$. Hence, the expected objective function for the few-shot diffusion models at a fixed time $t$ is:

$$\min_{\mathbf{s}_{V_{ta},\hat{\theta}} \in \tilde{\mathcal{Q}}_{NN}(\hat{\theta})} \mathcal{L}_{ta,t}(\mathbf{s}_{V_{ta},\hat{\theta}}) = \mathbb{E}_{X_{ta}\sim q_{ta}}\left[\ell_t^{ta}\left(X_{ta}; \mathbf{s}_{V_{ta},\hat{\theta}}\right)\right],$$

where

$$\tilde{\mathcal{Q}}_{NN}(\theta) = \left\{\mathbf{s}_{V,\theta}(X, t) = \frac{1}{\sigma_t^2}V\mathbf{f}_\theta\left(V^\top X, t\right) - \frac{1}{\sigma_t^2}X : V \in \mathbb{R}^{D \times d} \text{ with } \mathrm{rank}(V) = d.\right\},$$

Note that the constraint $\mathrm{rank}(V) = d$ is a weaker constraint than $V^\top V = I_d$ since $\mathrm{rank}(V) = d$ does not involve length information.

**Lemma C.2.** *Assume Assumption 3.1 and $q_z = \mathcal{N}(0, \lambda^2 I_d)$. Let $C = \mathbb{E}_{X_{ta}\sim q^{ta}}\left[X_{ta}X_{ta}^\top\right]$ be the expected data covariance matrix. Then, $\widetilde{V}_{ta}$ has a closed form:*

$$\widetilde{V}_{ta}\widetilde{V}_{ta}^\top = \frac{m_t^2\widehat{\lambda}^2 + \sigma_t^2}{\widehat{\lambda}^2}\left(C + \sigma_t^2 I_D\right)^{-1} C.$$

**Proof.** Let $\widehat{G}_t = I_d - \sigma_t^2\widehat{\Sigma}_t^{-1}$, then we have that

$$\ell_t^{ta}\left(X_{ta,i}; \mathbf{s}_{V_{ta},\hat{\theta}}\right) = \mathbb{E}_{X_t|X_0=X_{ta,i}}\left[\|\frac{1}{\sigma_t^2}V_{ta}\widehat{G}_t V_{ta}^\top X_t - \frac{1}{\sigma_t^2}X_t - \nabla\log q_t^{ta}(X_t|X_0)\|_2^2\right]$$

$$= \frac{1}{\sigma_t^4}\mathbb{E}_{X_t|X_0=X_{ta,i}}\left[\|V_{ta}\widehat{G}_t V_{ta}^\top X_t - m_t X_0\|_2^2\right]$$

where the second equality follows $\nabla\log q_t^{ta}(X_t|X_0) = -\frac{X_t - m_t X_0}{\sigma_t^2}$. Let $C = \mathbb{E}_{X_{ta}\sim q_{ta}}\left[X_{ta}X_{ta}^\top\right]$ be the expected covariance matrix of the target dataset. With the fact $\mathbb{E}_{X_t|X_0}[X_t X_t^\top] = m_t^2 X_0 X_0^\top + \sigma_t^2 I_D$ and $\mathbb{E}_{X_t|X_0}[X_0 X_t^\top] = m_t X_0 X_0^\top$, the optimization problem can be simplified to the following form (without misunderstanding, we ignore the subscript $ta$):

$$\min_{V \in \mathbb{R}^{D \times d}, \mathrm{rank}(V)=d} \mathcal{L}(V) = \|(m_t^2 C + \sigma_t^2 I_D)^{\frac{1}{2}}V\widehat{G}_t V^\top\|_F^2 - 2m_t^2\mathrm{tr}(V\widehat{G}_t V^\top C),$$

where $(m_t^2 C + \sigma_t^2 I_D)^{\frac{1}{2}}$ is meaningful since $(m_t^2 C + \sigma_t^2 I_D)$ is positive-definite matrix. Let $\widetilde{V}$ be the solution of the above minimization problem. We first ignore the constraint $\mathrm{rank}(V) = d$ and calculate $\partial\mathcal{L}(V)/\partial V = 0$ (since $\widetilde{V}$ also satisfied $\partial\mathcal{L}(V)/\partial V|_{V=\widetilde{V}} = 0$), we know that $\widetilde{V}$ satisfies the following equality:

$$(m_t^2 C + \sigma_t^2 I_D)V\widehat{G}_t V^\top V\widehat{G}_t = m_t^2 CV\widehat{G}_t,$$

which indicate

$$((m_t^2 C + \sigma_t^2 I_D)V\widehat{G}_t V^\top - m_t^2 C)(V\widehat{G}_t) = \mathrm{O}_{D \times d}.$$

The above equality means $\mathrm{rank}((m_t^2 C + \sigma_t^2 I_D)V\widehat{G}_t V^\top - m_t^2 C) + \mathrm{rank}(V\widehat{G}_t) \leq d$. Since $\mathrm{rank}(V) = d$ and $\mathrm{rank}(\widehat{G}_t) = d$, then we have that $\mathrm{rank}(V\widehat{G}_t) = d$ and

$$(m_t^2 C + \sigma_t^2 I_D)V\widehat{G}_t V^\top = m_t^2 C. \tag{5}$$

In Section 5, we assume the latent distribution is a isotropic Gaussian distribution $q_z = \mathcal{N}(0, \lambda^2 I_d)$. In this setting, $\widehat{\Sigma}$ is equal to $\widehat{\lambda}^2 I_d$ and $\widehat{G}_t = \frac{m_t^2\widehat{\lambda}^2}{m_t^2\widehat{\lambda}^2 + \sigma_t^2}$, which indicate the closed form solution of $\widetilde{V}$ is

$$\widetilde{V}\widetilde{V}^\top = \frac{m_t^2\widehat{\lambda}^2 + \sigma_t^2}{\widehat{\lambda}^2}(C + \sigma_t^2 I_D)^{-1}C.$$

The last step is to prove $\mathrm{rank}(\widetilde{V}\widetilde{V}^\top) = d$. Note that $\mathrm{rank}(C) = \mathrm{rank}\left(\mathbb{E}_{X_{ta}\sim q_{ta}}\left[X_{ta}X_{ta}^\top\right]\right) = \mathrm{rank}(A\Sigma A^\top) = d$, which indicates

$$\min\{\mathrm{rank}(\widetilde{V}\widetilde{V}^\top), \mathrm{rank}(m_t^2 C + \sigma_t^2 I_D)\} \geq d.$$

Combined with $\widetilde{V} \in \mathbb{R}^{D \times d}$, we complete the proof. ∎

## C.3 The Error Bound for the empirical Closed-form Solution

In this part, we prove the accuracy bound of the empirical version closed form solution $\widetilde{\widetilde{V}}\widetilde{\widetilde{V}}^\top$ w.r.t. $n_s$ and $n_{ta}$. The empirical solution has the following form (without misunderstanding, we ignore the subscript $ta$):

$$\widetilde{\widetilde{V}}\widetilde{\widetilde{V}}^\top = \frac{m_t^2\widehat{\lambda}^2 + \sigma_t^2}{\widehat{\lambda}^2}(m_t^2\bar{C} + \sigma_t^2 I_D)^{-1}\bar{C}\,,$$

where $\bar{C} = \frac{1}{n_{ta}}\sum_{i=1}^{n_{ta}} X_{ta,i}X_{ta,i}^\top$.

**Theorem 5.2.** *Assume Assumption 3.1 and $q_z = \mathcal{N}(0, \lambda^2 I_d)$. Let $\widehat{q}_z$ be the latent distribution generated by the pre-trained models with $(\widehat{V}_{ta}, \widehat{\Sigma})$ and $M = \frac{d^2\beta^2(d+\lambda^2)}{\lambda}\sqrt{Dd\log(Ddn_s)(d^2 \vee D)}$. Then, with probability $1 - \delta_1$, we have that for any $t \in [\delta, T]$*

$$\left\|\widetilde{\widetilde{V}}_{ta}\widetilde{\widetilde{V}}_{ta}^\top - A_{ta}A_{ta}^\top\right\|_F^2 \leq \widetilde{O}\left(\frac{d\log(\frac{1}{\delta_1})}{m_t^2\lambda^2 + \sigma_t^2}\left(\frac{M}{d\delta\sqrt{n_s}} + \frac{d}{n_{ta}}(m_t^2\lambda^2 + \sigma_t^2)^2\right)\right)\,.$$

**Proof.** The empirical solution indicates that

$$(m_t^2\bar{C} + \sigma_t^2 I_D)\widetilde{\widetilde{V}}\widetilde{\widetilde{V}}^\top = \frac{m_t^2\widehat{\lambda}^2 + \sigma_t^2}{\widehat{\lambda}^2}\bar{C}\,. \tag{6}$$

To analyze this equality, we first show that $\widehat{\lambda}^2$ is accurate enough. We know that $\mathrm{KL}\left(q_0\|\widehat{p}_T^{\mathrm{LD}}\right) = d\left(\lambda^2/\widehat{\lambda}^2 - \log(\lambda^2/\widehat{\lambda}^2) - 1\right)$. Let $M_1 = \frac{d^2\beta^2(d+\lambda_{\max}^2)}{\lambda_{\min}\delta}\sqrt{(d^2 + Dd)\log(Ddn_s)(d^2 \vee D)}$. Then, Lemma C.1 show that with probability $1 - \delta_1$, we have

$$\mathrm{KL}\left(q_0\|\widehat{p}_T^{\mathrm{LD}}\right) = d\left(\lambda^2/\widehat{\lambda}^2 - \log(\lambda^2/\widehat{\lambda}^2) - 1\right) \leq \widetilde{O}\left(M_1\sqrt{\frac{\log(1/\delta_1)}{n_s}}\right)\,.$$

Combined with the above inequality, we know that with probability $1 - \delta_1$, $|\lambda^2/\widehat{\lambda}^2 - 1| \leq \sqrt{\frac{M_1\sqrt{\log(1/\delta_1)}}{d\sqrt{n_s}}}$ by using the Taylor Expansion. By using Lemma D.2, we know that with probability $1 - \delta_1$:

$$\frac{m_t^2\widehat{\lambda}^2 + \sigma_t^2}{\widehat{\lambda}^2}\bar{C} \leq \widetilde{O}\left(\left(m_t^2\lambda^2 + \frac{\lambda^2\sigma_t^2}{\widehat{\lambda}^2}\right)\left(1 + \frac{2\sqrt{d + \log(1/\delta_1)}}{\sqrt{n_{ta}}}\right)AA^\top\right)$$

$$\leq \widetilde{O}\left(\left(m_t^2\lambda^2 + \sigma_t^2 + \sqrt{\frac{M_1\sqrt{\log(1/\delta_1)}}{d\sqrt{n_s}}}\sigma_t^2\right)\left(1 + \frac{2\sqrt{d + \log(1/\delta_1)}}{\sqrt{n_{ta}}}\right)AA^\top\right)\,. \tag{7}$$

For the left hand of Equation (6), we know that

$$(m_t^2\bar{C} + \sigma_t^2 I_D)\widetilde{\widetilde{V}}\widetilde{\widetilde{V}}^\top \geq \left(m_t^2\lambda^2 AA^\top - m_t^2\lambda^2\frac{2\sqrt{d + \log(1/\delta_1)}}{\sqrt{n_{ta}}} + \sigma_t^2 I_D\right)\widetilde{\widetilde{V}}\widetilde{\widetilde{V}}^\top \tag{8}$$

Combined with Equation (7) and Equation (8), we have that

$$\left(m_t^2\lambda^2 AA^\top + \sigma_t^2 I_D\right)\left(\widetilde{\widetilde{V}}\widetilde{\widetilde{V}}^\top - AA^\top\right)$$

$$\leq \widetilde{O}\left(\left(\sqrt{\frac{M_1\sqrt{\log(1/\delta_1)}}{d\sqrt{n_s}}} + \frac{2\sqrt{d + \log(1/\delta_1)}}{\sqrt{n_{ta}}}\left(m_t^2\lambda^2 + \sigma_t^2 + \sqrt{\frac{M_1\sqrt{\log(1/\delta_1)}}{d\sqrt{n_s}}}\sigma_t^2\right)\right)AA^\top\right.$$

$$\left. + 2m_t^2\lambda^2\frac{\sqrt{d + \log(1/\delta_1)}}{\sqrt{n_{ta}}}\widetilde{\widetilde{V}}\widetilde{\widetilde{V}}^\top\right)\,.$$

Let $M_2(n_s, n_{ta}) = \sqrt{\frac{M_1\sqrt{\log(1/\delta_1)}}{d\sqrt{n_s}}} + \frac{2\sqrt{d+\log(1/\delta_1)}}{\sqrt{n_{ta}}}\left(m_t^2\lambda^2 + \sigma_t^2 + \sqrt{\frac{M_1\sqrt{\log(1/\delta_1)}}{d\sqrt{n_s}}}\sigma_t^2\right)$. According-

ing to symmetry, we know that

$$\left\|(m_t^2\lambda^2 AA^\top + \sigma_t^2 I_D)\left(\bar{\tilde{V}}\bar{\tilde{V}}^\top - AA^\top\right)\right\|_F^2 \leq \tilde{O}\left(M_2(n_s, n_{ta})^2\|AA^\top\|_F^2 + \frac{m_t^4\lambda^4(d+\log(1/\delta_1))}{n_{ta}}\|\bar{\tilde{V}}\bar{\tilde{V}}^\top\|_F^2\right)$$

$$\leq \tilde{O}\left(D^2(M_2(n_s, n_{ta})^2)\right).$$

The last inequality follows that each element of $\bar{\tilde{V}}\bar{\tilde{V}}^\top$ is bounded by some constant due to the form of the empirical closed form solution and

$$\|AA^\top\|_F^2 = tr(AA^\top AA^\top) = tr(AA^\top) = tr(I_d) = d.$$

For the right hand of the above inequality, we know that

$$\left\|(m_t^2\lambda^2 AA^\top + \sigma_t^2 I_D)\left(\bar{\tilde{V}}\bar{\tilde{V}}^\top - AA^\top\right)\right\|_F^2$$

$$= \text{Tr}\left(\left(\bar{\tilde{V}}\bar{\tilde{V}}^\top - AA^\top\right)\left(\bar{\tilde{V}}\bar{\tilde{V}}^\top - AA^\top\right)(m_t^2\lambda^2 AA^\top + \sigma_t^2 I_D)(m_t^2\lambda^2 AA^\top + \sigma_t^2 I_D)\right)$$

$$\geq \left(m_t^2\lambda^2 + \sigma_t^2\right)\left\|\left(\bar{\tilde{V}}\bar{\tilde{V}}^\top - AA^\top\right)\right\|_F^2.$$

Then, we complete the proof ∎

## D   Auxiliary Lemmas

The following concentration lemma comes from Lemma 15 of Chen et al. (2023c).

**Lemma D.1** (Lemma 15, (Chen et al., 2023c)). *Let $\mathcal{G}$ be a bounded function class, i.e., there exists a constant $B$ such that any $g \in \mathcal{G} : \mathbb{R}^d \mapsto [0, B]$. Let $\mathbf{z}_1, \ldots, \mathbf{z}_n \in \mathbb{R}^d$ be i.i.d. random variables. For any $\delta \in (0, 1), a \leq 1$, and $\tau > 0$, we have*

$$\mathbb{P}\left(\sup_{g \in \mathcal{G}} \frac{1}{n}\sum_{i=1}^n g(\mathbf{z}_i) - (1+a)\mathbb{E}[g(\mathbf{z})] > \frac{(1+3/a)B}{3n}\log\frac{\mathcal{N}(\tau, \mathcal{G}, \|\cdot\|_\infty)}{\delta_1} + (2+a)\tau\right) \leq \delta_1 \quad \text{and}$$

$$\mathbb{P}\left(\sup_{g \in \mathcal{G}} \mathbb{E}[g(\mathbf{z})] - \frac{1+a}{n}\sum_{i=1}^n g(\mathbf{z}_i) > \frac{(1+6/a)B}{3n}\log\frac{\mathcal{N}(\tau, \mathcal{G}, \|\cdot\|_\infty)}{\delta_1} + (2+a)\tau\right) \leq \delta_1.$$

In the following lemma, we show the concentration of the data covariance matrix. Note that the proof sketch of the following lemma mainly follows Lemma 6 of (Du et al., 2020). We prove a concentration bound that depends on $n$ instead of a constant bound with a large enough $n$.

**Lemma D.2** (The Modified Lemma A.6, (Du et al., 2020)). *Let $\mathbf{a}_1, \ldots, \mathbf{a}_n$ be i.i.d. $d$-dimensional random vectors such that $\mathbb{E}[\mathbf{a}_i] = \mathbf{0}$, $\mathbb{E}[\mathbf{a}_i\mathbf{a}_i^\top] = I$, and $\mathbf{a}_i$ is $\rho^2$-subgaussian. Then with probability at least $1 - \delta_1$ we have*

$$\left(1 - \frac{2\rho^2\sqrt{d+\log(1/\delta_1)}}{\sqrt{n}}\right)I \preceq \frac{1}{n}\sum_{i=1}^n \mathbf{a}_i\mathbf{a}_i^\top \preceq \left(1 + \frac{2\rho^2\sqrt{d+\log(1/\delta_1)}}{\sqrt{n}}\right)I$$

**Proof.** Let $A = \frac{1}{n}\sum_{i=1}^n \mathbf{a}_i\mathbf{a}_i^\top - I$. Similar to Du et al. (2020), we use an $\epsilon$-net argument for the unit sphere $\mathcal{S}^{d-1} = \{\mathbf{v} \in \mathbb{R}^d : \|\mathbf{v}\| = 1\}$. For any $\mathbf{v} \in \mathcal{S}^{d-1}$, we know that $(\mathbf{v}^\top\mathbf{a}_i)^2 - 1$ is zero-mean and $16\rho^2$-subgaussian. By using the Bernstein inequality, we have for any $\epsilon > 0$

$$\Pr\left[|\mathbf{v}^\top A\mathbf{v}| > \epsilon\right] \leq 2\exp\left(-\frac{n}{2}\min\left\{\frac{\epsilon^2}{(16\rho^2)^2}, \frac{\epsilon}{16\rho^2}\right\}\right).$$

Next, we take a $\frac{1}{5}$-net $\mathcal{N} \subset \mathcal{S}^{d-1}$ of $\mathcal{S}^{d-1}$ with size $|\mathcal{N}| \le e^{O(d)}$. By using the union bound, we know that

$$\Pr\left[\max_{\boldsymbol{v} \in \mathcal{N}} \left|\boldsymbol{v}^\top A \boldsymbol{v}\right| > \epsilon\right] \le 2|\mathcal{N}| \exp\left(-\frac{n}{2}\min\left\{\frac{\epsilon^2}{(16\rho^2)^2}, \frac{\epsilon}{16\rho^2}\right\}\right)$$

$$\le \exp\left(O(d) - \frac{n}{2}\min\left\{\frac{\epsilon^2}{(16\rho^2)^2}, \frac{\epsilon}{16\rho^2}\right\}\right)$$

Let the right hand of the above inequality equals to $\delta_1$. We know that with probability $1 - \delta_1$:

$$\max_{\boldsymbol{v} \in \mathcal{N}} \left|\boldsymbol{v}^\top A \boldsymbol{v}\right| \le \frac{\rho^2 \sqrt{d + \log(1/\delta_1)}}{\sqrt{n}}.$$

Note that for any $\boldsymbol{u} \in \mathcal{S}^{d-1}$, there exists $\boldsymbol{u}' \in \mathcal{N}$ such that $\|\boldsymbol{u} - \boldsymbol{u}'\| \le \frac{1}{5}$. Then, we have that

$$|\boldsymbol{u}^\top A \boldsymbol{u}| \le |(\boldsymbol{u}')^\top A \boldsymbol{u}'| + 2|(\boldsymbol{u} - \boldsymbol{u}')^\top A \boldsymbol{u}'| + |(\boldsymbol{u} - \boldsymbol{u}')^\top A (\boldsymbol{u} - \boldsymbol{u}')|$$

$$\le \frac{\rho^2 \sqrt{d + \log(1/\delta_1)}}{\sqrt{n}} + \frac{1}{2}\|A\|_2,$$

where $\|A\|_2$ is the operator norm of matrix $A$. Taking a supreme over $u \in \mathcal{S}^{d-1}$, we have that

$$\|A\|_2 \le \frac{\rho^2 \sqrt{d + \log(1/\delta_1)}}{\sqrt{n}} + \frac{1}{2}\|A\|_2.$$

Then, we complete the proof. ∎

In Section 5, we assume the latent distribution is a Gaussian distribution instead of a subgaussian one. Yuan et al. (2023) show that in this setting, the approximation error bound has better dependence on $n_s$.

**Lemma D.3** (Lemma C.1, (Yuan et al., 2023)). *Assume the latent distribution is a Gaussian distribution $q_z = \mathcal{N}(0, \Sigma)$ with $\Sigma = \mathrm{diag}\left(\lambda_1^2, \ldots, \lambda_d^2\right) \succ 0$. Then, the solution $(\widehat{V}_s, \widehat{\theta})$ of Equation (1) has the following approximation error bound with probability $1 - \delta_1$:*

$$\frac{1}{T - \delta} \int_\delta^T \left\|\nabla \log q_t^s(\cdot) - \mathbf{s}_{\widehat{V}_s, \widehat{\theta}}(\cdot, t)\right\|_{L^2(q_t^s)}^2 \, dt \le O\left(\frac{1}{\delta}\sqrt{\frac{(d^2 + Dd)\log(Ddn_s)(d^2 \vee D)\log\frac{1}{\delta_1}}{n_s}}\right).$$

# E   Additional Experiments

In this section, we do experiments on real-world datasets to show that the new model obtained by only fine-tuning appropriate encoder and decoder layers on target datasets with only 10 images can generate novel images with the target dataset feature. On the contrary, if all parameters can be fine-tuned, the model will suffer from memory phenomenon and only generate the ten images in the target dataset. This phenomenon indicates that only fine-tuning the appropriate encoder and decoder will result in a model with a generalization property.

**Setting.**   In this experiment, we use a U-net network with attention layers, which contains 11 downblocks, 2 middleblocks, and 15 upblocks. When only fine-tuning the encoder and decoder layers, we fine-tune the first 4 downblock layers (encoder) and 4 upblock layers (decoder) instead of only using linear layers as the encoder and decoder (discuss in the later discussion paragraph).

The above experiments are conduct on a GeForce RTX 4090. We train the neural network using AdamW optimizer with learning rate 0.0001. For the pre-trained phase, we train the models for 200 epochs with batch size 20. It takes 5 hours to obtain a pre-trained diffusion models. For the fine-tuning phase, we fine-tune the pre-trained models for 400 epochs with batch size 2. It take 3 minutes to fine-tune the pre-trained models.

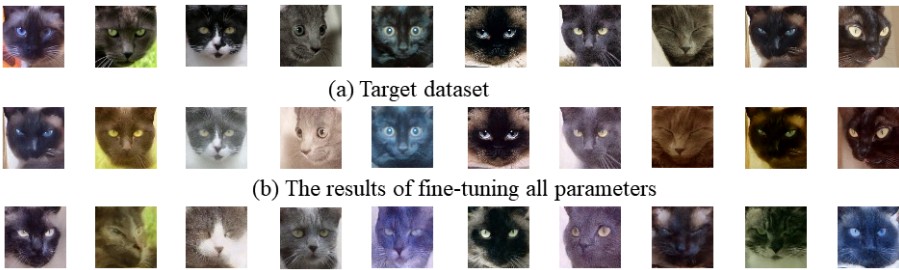

(a) Target dataset

(b) The results of fine-tuning all parameters

(c) The results of fine-tuning encoder and decoder

Figure 2: The experiments on cat face dataset

**Dataset.** Our experiments use 2 real-world datasets: the CelebA64 dataset and the cat face dataset.

- CelebA64 (size $3 * 64 * 64$).
  (a) Source dataset: 6400 images of faces with different hairstyles (without the bald feature).
  (b) Target dataset: 10 images with the bald feature in CelebA64.
- Cat face images (size $3 * 64 * 64$).
  (a) Source dataset: 4200 cat images with different colors (without black color cat).
  (b) Target dataset: 10 black color cat images (The color black constitutes more than 70% of the image's composition.).

**Discussion on results.** The experiment results of CelebA64 have been discussed in Section 6. The experiment phenomenon is similar for the cat face images, which means the models obtained by only fine-tuning the encoder and decoder can generate novel images with the target feature (Figure 2). We note that when choosing the target cat face dataset if the color black constitutes more than 70% of the image's composition, we view this cat image as the black cat. Hence, different colors exist for cats, such as white and grey, due to the target dataset containing a small number of these colors (such as images 1, 3, 4, 6, 8). As a result, our fine-tuning results also contain these colors. However, our results do not contain colors other than those in the target dataset and can produce novel samples, which also proves the effectiveness of our fine-tuning method.

**Discussion on linear encoder and decoder.** Assumption 3.1 assumes the linear subspace, which indicates linear encoder and decoder. However, we fine-tune the first 4 downblock layers (encoder) and 4 upblock layers (decoder) instead of only using linear layers as the encoder and decoder. We note that this operation does not conflict with our Assumption. Recall that in Stable Diffusion (Rombach et al., 2022), the diffusion models run in the VAE embedding space [7]. Hence, we can view the first 3 downblock layers and the last 3 upblock layers as the VAE encoder and VAE decoder. Then, we can obtain $X$ in this paper by running the VAE encoder. The remaining 1 downblock and 1 upblock layer can be viewed as linear encoder and decoder $A$. As mentioned in Section 4.2 of StyleGAN (Karras et al., 2019), the feature of $X$ obtained by running a good-enough VAE encoder has linear separability, which also supports our Assumption 3.1.

---

[7]To distinguish the latent space in this paper, we use embedding space here.

