# OpenReview forum: "Few-Shot Diffusion Models Escape the Curse of Dimensionality"
_NeurIPS.cc/2024/Conference — NeurIPS 2024 poster_

### Official Review · Reviewer_fbjU · 2024-07-08

**Soundness:** 4
**Presentation:** 3
**Contribution:** 3
**Rating:** 6
**Confidence:** 3

**Summary:**

The paper provides a theoretical analysis of the few-shot fine-tuning problem for diffusion models. It makes the following key assumptions: (1) the pretraining and fine-tuning data distributions share a common latent distribution, and (2) a specific network architecture. Under these assumptions, the paper proves that during the fine-tuning phase, the diffusion model avoids the curse of latent dimensionality and achieves an approximation bound of $\widetilde{\mathcal{O}}(n_s^{-2/d}+n_{ta}^{-1/2})$. Furthermore, when considering a low-rank Gaussian distribution, the model attains an accuracy bound of $\widetilde{\mathcal{O}}(1/n_{ta}+1/\sqrt{n_{s}})$. The paper also includes real-world experiments to support the theoretical findings.

**Strengths:**

The theoretical part of the paper is solid. As an extension of [1], the paper studies the diffusion model in the few-shot fine-tuning problem. Both the assumption of the data distribution and the definition of fine-tuning are reasonable to me. The conclusion aligns with the practice, that diffusion model fine-tuning escapes the curse of dimensionality and requires much fewer data samples to converge to the underlying distribution. I haven't gone through the details of the proof, if there is no further problem for them, I believe this is a theoretical solid paper.

[1] Chen, M., Huang, K., Zhao, T., and Wang, M. Score approximation, estimation and distribution recovery of diffusion models on low-dimensional data. arXiv preprint arXiv:2302.07194, 2023c.

**Weaknesses:**

There is a gap between the real-world experiments and the theoretical results. The experiments demonstrate two main points: (1) fine-tuning the entire diffusion model leads to memorization, and (2) fine-tuning only the encoder and decoder parts of the diffusion model will generalize. However, these findings only support the validity of the fine-tuning definition used in the paper.
The gap lies in two areas:
1. The paper lacks proof that fine-tuning the full model results in memorization.
2. The real-world experiments do not show that few-shot fine-tuning can escape the curse of dimensionality. Specifically, the diffusion model still requires a similar number of samples for fine-tuning datasets with different intrinsic dimensions.

I think the paper lacks sufficient real-world experiments to fully support its title, "Few-Shot Diffusion Models Escape the Curse of Dimensionality." With such an experiment, I believe the paper will be a strong submission.

**Questions:**

In section 4.1, Table 1, the requirement derived from $n_{ta}^{\frac{d+5}{4(1 - \alpha(n_s))}} \geq n_s$ is counterintuitive. For datasets like ImageNet with higher latent dimensions, the requirement of $n_{ta}$ is less. Could the author provide more discussions or give some real-world experiments to support this?

**Limitations:**

The paper has covered the limitations and societal impact.

---

> ### Author Rebuttal · Authors · 2024-08-07
>
> Thanks for your valuable comments and suggestions. We provide our response to each question below.
>
> **W1: The theoretical guarantee of the fully fine-tuned method.**
>
> As shown in our real experiment and [1], when fine-tuning all parameters with a small target dataset, models tend to overfit and lose the prior information from the pre-trained model. In our theorem, this phenomenon means that in the fine-tuning phase, the model does not use $\hat{\theta}$ learned by the pre-trained model and achieves a $n_{ta}^{-2/d}$ error bound, which **suffers from the curse of dimensionality**. From an intuitive perspective, the probability density function (PDF) of a distribution learned by an overfitting model is only positive at the interval around the target dataset, which is far away from the PDF of true distribution and leads to a large error term. We will add this proof and discussion in the next version.
>
> We also note that it is possible to avoid this phenomenon by using a specific loss [1] or carefully choosing the optimization epochs [2]. We leave them as interesting future works.
>
> **W2: The discussion on the curse of the dimensionality.**
>
> With a very limited target dataset (such as 5-10 images), if a method can fine-tune the pre-trained model (trained with a large source dataset) and generate novel images with the target feature, we say that this models escape the curse of dimensionality. As shown in W1, the fully fine-tuned method has a large approximation error (theoretical guarantee) and suffers the memory phenomenon (experiments). On the contrary, our method can escape the curse of dimensionality from the theoretical (Thm. 4.3) and empirical perspective.
>
> From the empirical perspective, as shown in Appendix E (dataset part), our experiments use **$6400$** source data (CelebA64) to train a pre-trained model. Then, we only use **$10$** target images to fine-tune this model and generate novel images with the target feature, which indicates our few-shot models escape the curse of dimensionality. For the cat dataset, we have a similar augmentation (4200 source data and $10$ target data). We note that $10$ target images are very limited compared to the source dataset (including CelebA64 and Cat dataset). Hence, our experiments support the theoretical results under each dataset.
>
> We will add the above discussion and description of the source and target dataset in Sec. 6 to avoid confusion. Thanks again for the concerns and valuable comments on our real-world experiments.
>
> **Q1: A clearer discussion of the results of Thm. 4.3 and Table 1.**
>
> We note that **the goal of the fine-tuning phase is to achieve the same order error bound compared with the pre-trained models**, which means that we consider the relative relationship between $n_{ta}$ and $n_s$. Hence, if the coefficient of $n_{ta}$ and $n_s$ has the same order, we can only consider $1/\sqrt{n_{ta}}$ and $n_s^{-\frac{2-2\alpha\left(n_s\right)}{d+5}}$. To support the above augmentation, we first recall the results and calculate the coefficients:
> $$
> \begin{align}
> \left(\frac{(1+\beta)^2 Dd^3}{\delta\left(T-\delta\right)\sqrt{n_{ta}}}+\frac{\left(d+C_Z\right) d^2 \beta^2}{\delta^2 c_0}n_s^{-\frac{2-2\alpha\left(n_s\right)}{d+5}}\right)\log\left(\frac{1}{\delta_1}\right).
> \end{align}
> $$
> The dominated term of coefficient for $n_{ta}$ and $n_s$ are $Dd^3/\delta$ and $\frac{d^3}{\delta^2c_0}$, respectively. The classic choice for the early stopping parameter $\delta$ and forward time $T$ are $10^{-3}$ and $10$, respectively [3]. Then, with $D=256\*256\*3$ as an example (Since smaller $D$ is more friendly to $n_{ta}$, our discussion holds for all datasets in Table 1.),  $Dd^3/\delta = d^3\*20\*10^6$ and $\frac{d^3}{\delta^2c_0} = d^3\*10^6/c_0$, which has the same order. Hence, we consider the relative relationship between $1/\sqrt{n_{ta}}$ and $n_s^{-\frac{2-2\alpha\left(n_s\right)}{d+5}}$ and require
> $$
> \frac{1}{\sqrt{n_{ta}}}\leq n_s^{-\frac{2-2\alpha\left(n_s\right)}{d+5}},
> $$
> which indicates $n_{ta}^{\frac{d+5}{4(1-\alpha(n_s))}}\ge n_s$.
>
> We now discuss the reason why the requirement on $n_{ta}$ is smaller when $d$ is larger. We note that the error bound of the pre-trained model is heavily influenced by the latent dimension $d$. More specifically, when $d$ is large (e.g. ImageNet), the error bound of pre-trained models has a large error even with large-size source data. We only need a few target data to achieve the same error. When $d$ is small (e.g. CIFAR-10), pre-trained models have a small error, and we need a slightly large target data size to achieve a small error.
>
> We will add the above discussion in our next version to make it clearer.
>
> [1] Ruiz, N., Li, Y., Jampani, V., Pritch, Y., Rubinstein, M., & Aberman, K.  Dreambooth: Fine tuning text-to-image diffusion models for subject-driven generation. CVPR 2023.
>
> [2] Li, P., Li, Z., Zhang, H., & Bian, J. (2024). On the generalization properties of diffusion models. *Advances in Neural Information Processing Systems*, *36*.
>
> [3] Karras, T., Aittala, M., Aila, T., & Laine, S. (2022). Elucidating the design space of diffusion-based generative models. *Advances in neural information processing systems*, *35*, 26565-26577.

---

> > ### Comment · Reviewer_fbjU · 2024-08-11
> >
> > Thanks for your careful response. For W1, I agree the theory does show with full model fine-tuning, there will be a large error bound. However, a large error bound is not equivalent to the memorization phenomenon. Therefore, I think to support this part of the experiment, the author still needs some theoretical analysis on memorization. I am satisfied with the other answers. Overall, I think the work is valuable, I will increase the score to 6.

---

> > > ### Author Response · Authors · 2024-08-11
> > >
> > > Thank you for your positive feedback and support! We will add the theoretical guarantee for the fully fine-tuned method in the main content. For the explanation of the memorization phenomenon, we will add a detailed discussion in the future work paragraph according to your comments.

---

### Official Review · Reviewer_7GpV · 2024-07-11

**Soundness:** 3
**Presentation:** 3
**Contribution:** 3
**Rating:** 7
**Confidence:** 2

**Summary:**

The paper provides a theoretical analysis showing that fine-tuning pretrained diffusion models on a few target samples achieves a small approximation error, particularly in that it requires much fewer target data than source data. The authors also show that the solution has a closed form in a special case, demonstrating the ease of optimization for few-shot models. They provide bounds for this closed-form solution, which also shows better dependency on the target sample size than on the source sample size. Experiments on CelebA are conducted to further support the theoretical conclusions.

**Strengths:**

1. This work offers a theoretical understanding of why few-shot diffusion models can achieve strong performance despite the curse of dimensionality, which is a significant and intriguing topic.
2. The conclusions are well-supported by theoretical results and are clearly explained. Section 4 seems solid and well-structured, although I have not examined the proof in the appendix. Additionally, in Section 5, they demonstrate the closed-form expression of the minimizer under an isotropic Gaussian latent distribution and its ability to recover the true subspace, which further adds comprehensiveness and aids illustration.

**Weaknesses:**

I don't have any particular criticisms; however, I am not an expert in this topic, particularly regarding the theoretical analysis of diffusion models. I would like to hear other reviewers' opinions.

**Questions:**

I wonder if there is any intuitive explanation for assumptions like 4.1 and 4.2, particularly what they could mean in practice. Or are they just technical assumptions without much intuition?

**Limitations:**

The authors have discussed certain limitations in Section 7.

---

> ### Author Rebuttal · Authors · 2024-08-07
>
> Thanks for your valuable comments and suggestions on our assumptions. We discuss each assumption in detail below and will make it clearer in our next version.
>
> **Q1: The discussion on each assumption.**
>
> (a) The linear subspace and shared latent assumption. (Assumption 3.1).
>
> Since the diffusion models can find and adapt to the low-dimensional manifold [1], many theoretical works assume the linear low-dimensional manifold as the first step to understanding this phenomenon [2] [3], and we make exactly the same assumption compared with them.
>
> For the analysis of few-shot learning, it is a standard, natural, and necessary assumption to assume the shared latent representation [4].
>
> (a) The subgaussian latent variable (Assumption 4.1).
>
> In this assumption, we assume the latent variable is a subgaussian variable. Since a bounded variable is a subgaussian one, this assumption is naturally satisfied by the real-world image datasets. We note that this assumption is more realistic than the Gaussian assumption and is widely used in many theoretical works of diffusion models [1] [2].
>
> (b) The $\beta$-Lipschitz assumption on the score function (Assumption 4.2).
>
> We note that the $\beta$-Lipschitz assumption is a standard assumption on the score function for the theoretical works [2] [5]. As shown in Sec. 3.2 of [5], for a subgaussian variable (Assumption 4.1), we can replace $\beta$ with  $C_Z/\delta^2$ and remove this assumption. We will add the above discussion to our paper.
>
>
>
> [1] Tang, R., & Yang, Y. (2024, April). Adaptivity of diffusion models to manifold structures. In *International Conference on Artificial Intelligence and Statistics* (pp. 1648-1656). PMLR.
>
> [2] Chen, M., Huang, K., Zhao, T., & Wang, M. (2023, July). Score approximation, estimation and distribution recovery of diffusion models on low-dimensional data. In *International Conference on Machine Learning* (pp. 4672-4712). PMLR.
>
> [3] Yuan, H., Huang, K., Ni, C., Chen, M., & Wang, M. (2024). Reward-directed conditional diffusion: Provable distribution estimation and reward improvement. *Advances in Neural Information Processing Systems*, *36*.
>
> [4] Chua, K., Lei, Q., & Lee, J. D. (2021). How fine-tuning allows for effective meta-learning. *Advances in Neural Information Processing Systems*, *34*, 8871-8884.
>
> [5] Chen, S., Chewi, S., Li, J., Li, Y., Salim, A., & Zhang, A. R. (2022). Sampling is as easy as learning the score: theory for diffusion models with minimal data assumptions. *arXiv preprint arXiv:2209.11215*.

---

### Official Review · Reviewer_kXcP · 2024-07-11

**Soundness:** 3
**Presentation:** 3
**Contribution:** 3
**Rating:** 7
**Confidence:** 4

**Summary:**

This paper provides new bounds for the score function approximation and optimization in fine-tuning pretrained diffusion models (under some simplifying assumptions including linear structure of data distributions and equal latent distributions). The new approximation error bound depends only on the square root of the finetuning target sample size, which in turn suggests that fine-tuning diffusion models does not suffer from the curse of dimensionality (unlike their pretraining error bound which has an exponential dependence on the latent dimension). Furthermore, the accuracy bound and closed-form minimizer provides interesting insights into the fine-tuning behavior of diffusion models. The paper also provides limited empirical support for its theoretical claims on CelebA.

**Strengths:**

The paper provides novel theoretical bounds that can explain an important property of diffusion models (namely their fine-tuning convergence). Aside from the inherent theoretical value, I also think the provided insights can be very valuable both for the current practice of fine-tuning diffusion models, and for their future advancement. The paper is well organized and mostly easy to read (although the paper can benefit from simplifying its notations a bit more).

**Weaknesses:**

I do not see any major weakness in this work, but I do have two minor concerns:

1- The paper mentions that data-efficient fine-tuning of diffusion models is very successful and tries to explain this phenomenon. However, the paper does not clarify the extent of this success. I think the paper would benefit from discussing the many ways in which existing fine-tuning methods are suboptimal (diversity, out of distribution, etc), so as to provide a more nuanced context to the reader.

2- The paper does not sufficiently discuss the importance of its assumption, that is, what happens if each one breaks? Which ones are more crucial than others? Which ones are less realistic than others?

**Questions:**

Please see my questions in the Weaknesses section.

**Limitations:**

The paper sufficiently discusses its weaknesses.

---

> ### Author Rebuttal · Authors · 2024-08-07
>
> Thanks for your valuable comments and suggestions. We provide our response to each question below.
>
> **W1: The discussion on the existing fine-tuning method.**
>
> In this part, we discuss the fully fine-tuned method and explain why we need data-efficient fine-tuning methods.
>
> In earlier times, fully fine-tuned methods, such as DreamBooth [1], provided an important boost for developing few-shot models. However, they also show that the diffusion models suffer from the overfitting and memory phenomenon when fine-tuning all parameters (Fig. 6 of their paper). To deal with this problem, DreamBooth designs a prior preservation loss as a regularizer, which is needed carefully designed. Furthermore, a fully fine-tuned method is both memory and time inefficient [2].
>
> To avoid the above problem, many works fine-tune some key parameters and achieve success in many areas such as text2image and medical image generation [3] [4]. These works not only preserve the prior information but also have a significantly smaller model size, which is more practical for real-world applications. Hence, we focus on these models in our work.
>
> We will add the above discussion to our introduction part.
>
> **W2: The discussion on each assumption.**
>
> We note that Assumption 3.1 is the most important assumption for our analysis, and we discuss this assumption in detail below. The other assumptions are standard, and we discuss them one by one.
>
> (a) The linear subspace and shared latent assumption. (Assumption 3.1).
>
> Since the diffusion models can find and adapt to the low-dimensional manifold [5], many theoretical works assume the linear low-dimensional manifold as the first step to understand this phenomenon [6] [7] and we do exactly the same assumption compared with them. We note that since our analysis depends on the formula of score function under the linear subspace assumption, this assumption is important for our paper. An interesting future works is to extend our analysis to the nonlinear low-dimensional manifold and we will discuss it in our future work part.
>
> For the analysis of few-shot learning, it is a standard, natural and necessary assumption to assume the shared latent representation [8] [9].
>
> (b) The subgaussian latent variable and $\beta$-Lipschitz score function (Assumption 4.1 and 4.2).
>
> Since a bounded variable is a subgaussian one, this assumption is naturally satisfied by the real-world image datasets and is widely used in many theoretical works [6] [7].  For the Lipschitz score, this assumption is common for the theoretical works of diffusion models [6] [10].
>
> (c) The isotropic Gaussian latent variable assumption (Sec. 5).
>
> When considering the few-shot optimization problem, we assume the latent variable is isotropic Gaussian, which is stronger than subgaussian one in Assumption 4.1. We note that this assumption is necessary for the closed form solution in Lem. 5.1. However, as discussed in future work paragraph, it is not necessary if we use some optimization algorithm instead of directly obtaining a closed-form.
>
> Thanks for the concerns on our assumption. We will discuss our assumption in detail in the next version.
>
>
>
> [1] Ruiz, N., Li, Y., Jampani, V., Pritch, Y., Rubinstein, M., & Aberman, K.  Dreambooth: Fine tuning text-to-image diffusion models for subject-driven generation. CVPR 2023.
>
> [2] Xiang, C., Bao, F., Li, C., Su, H., & Zhu, J. (2023). A closer look at parameter-efficient tuning in diffusion models. *arXiv preprint arXiv:2303.18181*.
>
> [3] Han, L., Li, Y., Zhang, H., Milanfar, P., Metaxas, D., & Yang, F. Svdiff: Compact parameter space for diffusion fine-tuning. ICCV 2023.
>
> [4] Dutt, R., Ericsson, L., Sanchez, P., Tsaftaris, S. A., & Hospedales, T. Parameter-efficient fine-tuning for medical image analysis: The missed opportunity. *arXiv preprint arXiv:2305.08252*.
>
> [5] Tang, R., & Yang, Y. (2024, April). Adaptivity of diffusion models to manifold structures. In *International Conference on Artificial Intelligence and Statistics* (pp. 1648-1656). PMLR.
>
> [6] Chen, M., Huang, K., Zhao, T., & Wang, M. (2023, July). Score approximation, estimation and distribution recovery of diffusion models on low-dimensional data. In *International Conference on Machine Learning* (pp. 4672-4712). PMLR.
>
> [7] Yuan, H., Huang, K., Ni, C., Chen, M., & Wang, M. (2024). Reward-directed conditional diffusion: Provable distribution estimation and reward improvement. *Advances in Neural Information Processing Systems*, *36*.
>
> [8] Du, S. S., Hu, W., Kakade, S. M., Lee, J. D., & Lei, Q. (2020). Few-shot learning via learning the representation, provably. *arXiv preprint arXiv:2002.09434*.
>
> [9] Chua, K., Lei, Q., & Lee, J. D. (2021). How fine-tuning allows for effective meta-learning. *Advances in Neural Information Processing Systems*, *34*, 8871-8884.
>
> [10] Chen, S., Chewi, S., Li, J., Li, Y., Salim, A., & Zhang, A. R. (2022). Sampling is as easy as learning the score: theory for diffusion models with minimal data assumptions. *arXiv preprint arXiv:2209.11215*.

---

> > ### Comment · Reviewer_kXcP · 2024-08-09
> >
> > Thank you for your response, and the additional discussion of your assumptions. I keep my rating as I think this is a very valuable work.

---

> > > ### Author Response · Authors · 2024-08-10
> > >
> > > Thank you for your positive feedback and support!  We will add the discussion on the existing fine-tuning methods and our assumption according to your comments. In case you have any other questions, please don't hesitate to let us know.

---

### Official Review · Reviewer_XEHa · 2024-07-12

**Soundness:** 2
**Presentation:** 2
**Contribution:** 2
**Rating:** 4
**Confidence:** 4

**Summary:**

This paper studies the few-shot transfer in diffusion models. Specifically, it focuses on bounding the score-matching loss for the target distribution, which can be considered the estimation error of the corresponding score function. By assuming the source and target distribution share the same linear structure distribution on latent space of dimension $d$,  it gives a $\tilde{\mathcal{O}}(n_s^{-2/d} + n_{ta}^{-1/2})$ bound for training the few-shot diffusion model with $n_s$ source domain data and $n_{ta}$ target domain data. As claimed by the authors, this breaks the curse of dimensionality for the target domain with an improvement from $n_{ta}^{-2/d}$ (no few-shot adaptation) to $n_{ta}^{-1/2}$. In addition, they also proved a bound $\tilde{\mathcal{O}}(2/n_{ta} + 1/\sqrt{n_s})$ for the optimization error of score matching loss for a latent Gaussian special case.

**Strengths:**

* This paper provides theoretical guarantees for few-shot diffusion, illustrating sample complexity improvement w.r.t non-transfer diffusion process.
* The structure is clear and easy to understand, even though the writing quality could be further improved.

**Weaknesses:**

* Approximation error has a specific definition in learning theory, which characterizes the inductive bias caused by the selection of the hypothesis class [1]. When using the terminology as another definition, e.g., the score matching loss of an estimated score function with early stopping in Theorem 4.3, it should be obviously defined to avoid confusion.
* I have the following concerns about presenting and interpreting the bounds:
	* In Theorem 4.3, the bound depends on $F$, which is related to $d^3$. So, the order should be $\tilde{\mathcal{O}}(d^3(n_s^{-2/d} + n_{ta}^{-1/2}))$, I am not sure if $d^3$ could be hidden directly.
	* According to this, when $d$ becomes large, the error should grow, which means the requirement of $n_{ta}$ should grow. However, in Table 1, with the increase of latent dimension, $n_{ta}$ decreases for ImageNet and MS-COCO. This is contradictory to your results.
	* When $d<4$, isn't few-shot fine-tuning worse than directly training given the theorem?

* I am not quite convinced by the explanation of the experiments.
	* How is the difference between fine-tuning all the parameters and fine-tuning a smaller fraction of encoder-decoders reflected on your bounds?
	* I think there exist many ways to adjust the hyper-parameter for fine-tuning all the parameters to achieve similar results as fine-tuning a small part.

[1] Shalev-Shwartz, Shai, and Shai Ben-David. _Understanding machine learning: From theory to algorithms_. Cambridge University Press, 2014.

**Questions:**

1. Why $\tilde{V}_{ta}$ in Lemma 5.1 depends on time $t$ ? Shouldn't it be the minimizer of the integral over $t$?


Grammar: Line 271, 278, 280, 289, etc ...
I suggest the authors check the grammar and revise the paper accordingly.

**Limitations:**

Yes.

---

> ### Author Rebuttal · Authors · 2024-08-07
>
> Thanks for your valuable comments and suggestions. We provide our response to each question below.
>
> **W1 & Presentation.**
>
> Thanks again for the valuable comments. Before using the approximation error to represent the score matching loss with finite datasets, we will make a clear definition in the notation paragraph to avoid confusion. We will also polish our presentation according to your suggestions.
>
> **W2 (a) A clearer discussion of the results of Thm. 4.3.**
>
> We note that the goal of the fine-tuning phase is to achieve the same order error bound compared with the pre-trained models, which means that we consider the relative relationship between $n_{ta}$ and $n_s$. Hence, if the coefficient of $n_{ta}$ and $n_s$ has the same order, we can only consider $1/\sqrt{n_{ta}}$ and $n_s^{-\frac{2-2\alpha\left(n_s\right)}{d+5}}$. To support the above augmentation, we first recall the results and calculate the coefficients:
> $$
> \begin{align}
> \left(\frac{(1+\beta)^2 Dd^3}{\delta\left(T-\delta\right)\sqrt{n_{ta}}}+\frac{\left(d+C_Z\right) d^2 \beta^2}{\delta^2 c_0}n_s^{-\frac{2-2\alpha\left(n_s\right)}{d+5}}\right)\log\left(\frac{1}{\delta_1}\right).
> \end{align}
> $$
> The dominated term of coefficient for $n_{ta}$ and $n_s$ are $Dd^3/\delta$ and $\frac{d^3}{\delta^2c_0}$, respectively. The classic choice for the early stopping parameter $\delta$ and forward time $T$ are $10^{-3}$ and $10$, respectively [1]. Then, with $D=256\*256\*3$ as an example (Since smaller $D$ is more friendly to $n_{ta}$, our discussion holds for all datasets in Table 1.),  $Dd^3/\delta = d^3\*20\*10^6$ and $\frac{d^3}{\delta^2c_0} = d^3\*10^6/c_0$, which has the same order. Hence, we consider the relative relationship between $1/\sqrt{n_{ta}}$ and $n_s^{-\frac{2-2\alpha\left(n_s\right)}{d+5}}$. We will add the above discussion and provide a clearer explanation.
>
> **W2 (b) The discussion of Table 1.**
>
> As shown in W2 (a), the fine-tuning phase aims to achieve the same error bound compared with source data. To achieve this goal, the first step is to determine the error bound of the pre-trained model. We note that this error bound is heavily influenced by the latent dimension $d$. More specifically, when $d$ is large (e.g. ImageNet), the error bound of pre-trained models has a large error even with large-size source data. We only need a few target data to achieve the same error. When $d$ is small (e.g. CIFAR-10), pre-trained models have a small error, and we need a slightly large target data size to achieve a small error.
>
> **W2 (c): Directly training for small $d$.**
>
> For $d\leq 4$, the models achieve $1/\sqrt{n_{ta}}$ without the prior information from the source data. However, $d$ of common image datasets is larger than $20$ (Table 1). Hence, it is meaningful to consider our few-shot fine-tuning process, which fully uses prior information.
>
> **W3 (b): The discussion on fully fine-tuned methods.**
>
> In earlier times, fully fine-tuned methods, such as DreamBooth [2], provided an important boost for developing few-shot models. However, they also show that the diffusion models suffer from the overfitting and memory phenomenon when fine-tuning all parameters (Fig. 6 of their paper). To deal with this problem, they design a prior preservation loss as a regularizer, which needs to be carefully designed. Furthermore, a fully fine-tuned method is both memory and time inefficient [6].
>
> To avoid the above problem, many works fine-tune some key parameters and achieve success in many areas, such as text2image and medical image generation [3] [4]. These works not only preserve the prior information but also have a significantly smaller model size, which is more practical for applications. Hence, we focus on these models in our work. We will add the above discussion to our introduction.
>
> **W3 (a): The bound for fully fine-tuned methods.**
>
> As discussed above, the fully fine-tuned method tend to overfit and lose the prior information from the pre-trained model. In our theorem, this phenomenon means that in the fine-tuning phase, the model does not use $\hat{\theta}$ learned by the pre-trained model and achieves a $n_{ta}^{-2/d}$ error bound, which suffers from the curse of dimensionality. From an intuitive perspective, the probability density function (PDF) of a distribution learned by an overfitting model is only positive at the interval around the target dataset, which is far away from the PDF of true distribution and leads to a large error term.
>
> We also note that it is possible to avoid this phenomenon by using a specific loss [2] or carefully choosing the optimization epochs [5]. We leave them as interesting future works.
>
> **Q1: The optimization problem with a fixed $t$.**
>
> When considering a linear subspace, the diffusion process happens at the latent space, and $A_{ta}$ is independent of $t$. Hence, we fix a $t$ to solve the optimization problem. As discussed in Sec. 5.1, the objective is more flexible than traditional PCA since it can choose suitable $t$ to avoid the influence of large $\lambda^2$.
>
> [1] Karras, T., Aittala, M., Aila, T., & Laine, S. Elucidating the design space of diffusion-based generative models. NeurIPS 2022.
>
> [2] Ruiz, N., Li, Y., Jampani, V., Pritch, Y., Rubinstein, M., & Aberman, K.  Dreambooth: Fine tuning text-to-image diffusion models for subject-driven generation. CVPR 2023.
>
> [3] Han, L., Li, Y., Zhang, H., Milanfar, P., Metaxas, D., & Yang, F. Svdiff: Compact parameter space for diffusion fine-tuning. ICCV 2023.
>
> [4] Dutt, R., Ericsson, L., Sanchez, P., Tsaftaris, S. A., & Hospedales, T. Parameter-efficient fine-tuning for medical image analysis: The missed opportunity. *arXiv preprint arXiv:2305.08252*.
>
> [5] Li, P., Li, Z., Zhang, H., & Bian, J. On the generalization properties of diffusion models. NeurIPS 2023.
>
> [6] Xiang, C., Bao, F., Li, C., Su, H., & Zhu, J. (2023). A closer look at parameter-efficient tuning in diffusion models. *arXiv preprint arXiv:2303.18181*.

---

> > ### Comment · Reviewer_XEHa · 2024-08-13
> >
> > Thanks for your response. I am not fully convinced by your reply. So I will not change my score at current stage. I need more time to recall the paper's details and verify your response. I will adjust my score based on your response and discussion with other reviewers.

---

> > > ### Author Response · Authors · 2024-08-13
> > >
> > > We would like to thank the reviewer again for the time and efforts! We are more than happy to discuss our work in the rebuttal phase. In case you have any other questions and concerns, please don't hesitate to let us know.
> > >
> > > Best regards
> > >
> > > Author

---

### Comment · Area_Chair_HnNn · 2024-08-11
**Reviewer-Author discussion period**

Dear Reviewers,

The deadline for the reviewer-author discussion period is approaching. If you haven't done so already, please review the rebuttal and provide your response at your earliest convenience.

Best wishes, AC

---

### Decision · Program_Chairs · 2024-09-25

**Decision:**

Accept (poster)

**Comment:**

The reviewers agree that the paper presents a strong and well-founded theoretical framework for understanding few-shot diffusion models. As highlighted by Reviewer kXcP, the theory is both novel and interesting, offering meaningful insights into the training of few-shot diffusion models. Furthermore, the theoretical contributions are robustly supported by comprehensive experimental results, enhancing the overall impact and validity of the work.